# Neuropsychological evidence of multi-domain network hubs in the human thalamus

Kai Hwang[1,2,3,4]*, James M Shine[5], Joel Bruss[3,6,7], Daniel Tranel[1,3,6], Aaron Boes[3,4,6,7]

[1]Department of Psychological and Brain Sciences, The University of Iowa & The University of Iowa College of Medicine, Iowa City, United States; [2]Cognitive Control Collaborative, The University of Iowa & The University of Iowa College of Medicine, Iowa City, United States; [3]Iowa Neuroscience Institute, The University of Iowa & The University of Iowa College of Medicine, Iowa City, United States; [4]Department of Psychiatry, The University of Iowa & The University of Iowa College of Medicine, Iowa City, United States; [5]Brain and Mind Center, The University of Sydney, Sydney, Australia; [6]Department of Neurology, The University of Iowa & The University of Iowa College of Medicine, Iowa City, United States; [7]Department of Pediatrics, The University of Iowa & The University of Iowa College of Medicine, Iowa City, United States

**Abstract** Hubs in the human brain support behaviors that arise from brain network interactions. Previous studies have identified hub regions in the human thalamus that are connected with multiple functional networks. However, the behavioral significance of thalamic hubs has yet to be established. Our framework predicts that thalamic subregions with strong hub properties are broadly involved in functions across multiple cognitive domains. To test this prediction, we studied human patients with focal thalamic lesions in conjunction with network analyses of the human thalamocortical functional connectome. In support of our prediction, lesions to thalamic subregions with stronger hub properties were associated with widespread deficits in executive, language, and memory functions, whereas lesions to thalamic subregions with weaker hub properties were associated with more limited deficits. These results highlight how a large-scale network model can broaden our understanding of thalamic function for human cognition.

*For correspondence:
kai-hwang@uiowa.edu

Competing interest: The authors declare that no competing interests exist.

## Introduction

Hubs are highly connected network components crucial for network functions. In the human brain, hubs are thought to facilitate communication between information-processing systems, and support cognitive functions that likely arise from brain-wide network interactions (*Bertolero et al., 2015*; *Gratton et al., 2018*; *van den Heuvel and Sporns, 2013*). Prior studies have identified hubs in frontoparietal (FP) association cortices that have extensive connections with distributed brain regions (*Goldman-Rakic, 1988*; *Hagmann et al., 2008*; *Power et al., 2013*). These FP hubs are behaviorally significant. For example, the connectivity pattern of cortical hubs correlates with behavioral task performance (*Bertolero et al., 2018*; *Cole et al., 2013*), and focal damage to cortical hubs is associated with behavior across cognitive domains (*Reber et al., 2021*; *Warren et al., 2014*).

The human thalamus also possesses hub-like network properties (*Cole et al., 2010*; *Hwang et al., 2017*). The thalamus consists of different constituent subregions, each with a unique functional, anatomical, and connectivity profile (*Sherman and Guillery, 2013*). Many thalamic subregions exhibit 'many-to-one' and 'one-to-many' connectivity motifs—a subregion receives converging projections

from multiple cortical regions, and simultaneously projects to multiple cortical regions (*Giguere and Goldman-Rakic, 1988*; *Guillery and Sherman, 2002*; *Selemon and Goldman-Rakic, 1988*). This connectivity motif is a hallmark characteristic of higher-order thalamic nuclei (i.e., the mediodorsal nucleus [MD]), and can be supported by a specific group of thalamocortical projection cells, the 'matrix' cells (*Jones, 2009*). Matrix cells have a diffuse and distributed projection pattern; they innervate the superficial layers of multiple cortical regions, crossing receptive fields and functional boundaries. Consistent with these anatomical features, formal network analyses have found 'connector hubs' in the human thalamus. Thalamic connector hubs have strong functional connectivity with multiple cortical networks (*Greene et al., 2020*; *Hwang et al., 2017*), and each network can be associated with a distinct set of cognitive functions (*Bertolero et al., 2015*; *Crossley et al., 2013*; *Yeo et al., 2015*). This connectivity architecture suggests that a thalamic hub participates in functions that involve multiple networks and might contribute to many different cognitive functions.

However, the behavioral significance of this important network position of the thalamus has yet to be established. Large-scale meta-analyses of functional neuroimaging research, primarily functional magnetic resonance imaging (fMRI) studies, have found that tasks from many different cognitive domains (e.g., executive function, memory, perception) are associated with increased activity in overlapping thalamic subregions (*Hwang et al., 2017*; *Yeo et al., 2015*). Functional neuroimaging findings are nevertheless correlational, because increased blood-oxygen-level-dependent (BOLD) activity in response to a particular behavior is not evidence that this brain region is necessary for the studied behavior (*Sutterer and Tranel, 2017*). The lesion method, studying patients with focal damage to the thalamus, can provide a stronger test of whether thalamic hubs are necessary for human cognition.

Based on the prominent hub property of the thalamus, we hypothesize that thalamic hubs are involved in multi-domain processing involving multiple functional systems and contribute to behavior across multiple cognitive domains. To test this hypothesis, we combined neuropsychological evaluations from patients with focal thalamic lesions with network analyses of the human thalamocortical functional connectome. To evaluate behavior across cognitive domains, we analyzed neuropsychological tests that assess executive, language, memory, learning, visuospatial, and construction functions. To evaluate the hub properties of different thalamic subregions, we used two different measures: first, a well-established graph-theoretic metric, participation coefficient (PC), and second, the estimated density of matrix projection cells. PC estimates the distribution of coupling between a brain region with multiple brain systems in the functional connectome (*Power et al., 2013*; *Warren et al., 2014*). Hub regions with higher PC values are thought to participate in processes recruiting multiple functional networks (*Shine et al., 2016*). Matrix cells in the thalamus diffusely project to multiple brain regions, and thus thalamic hub regions should have a higher density of matrix cells. We predict that patients with focal lesions to thalamic subregions with high PC values and high density of matrix cells will exhibit more extensive impairment across cognitive domains. In contrast, lesions to thalamic subregions with lower PC values and lower density of matrix cells will exhibit more limited impairment.

## Results

We identified 20 patients (ages 18–70, 13 males) with focal lesions restricted to the thalamus, and 21 comparison patients (ages 19–77 years, 8 males) with lesions that spared the thalamus (*Figure 1A*). All patients were drawn from the Iowa Neurological Patient Registry. The registry contains data from patients with focal, stable brain lesions who have undergone neuropsychological assessment and brain imaging at least 3 months after lesion onset. Comparison patients had lesions that spared the thalamus, predominately lesioned the gray matter (GM) (>80 % of total lesion volume), but were similar in size to those found in thalamic patients (≤5088 mm³, size of the largest observed thalamic lesion). Lesion sizes were not significantly different between groups (thalamus group: mean = 1364 mm³, SD = 1212 mm³; comparison group: mean = 1334 mm³, SD = 732 mm³; group difference in lesion size, p = 0.31). There were no significant group differences in age, full scale IQ, verbal IQ, and performance IQ between thalamus patients and comparison patients.

We predicted that lesions to hub regions in the thalamus will affect behavior across different cognitive domains. To test the behavioral relevance of the thalamic hubs, we first compared neuropsychological outcomes between the thalamus and comparison patients. The goal of this comparison was to determine whether lesions of the thalamus were associated with cognitive impairment, beyond any nonspecific lesion effects. We analyzed outcome data from several neuropsychological tests,

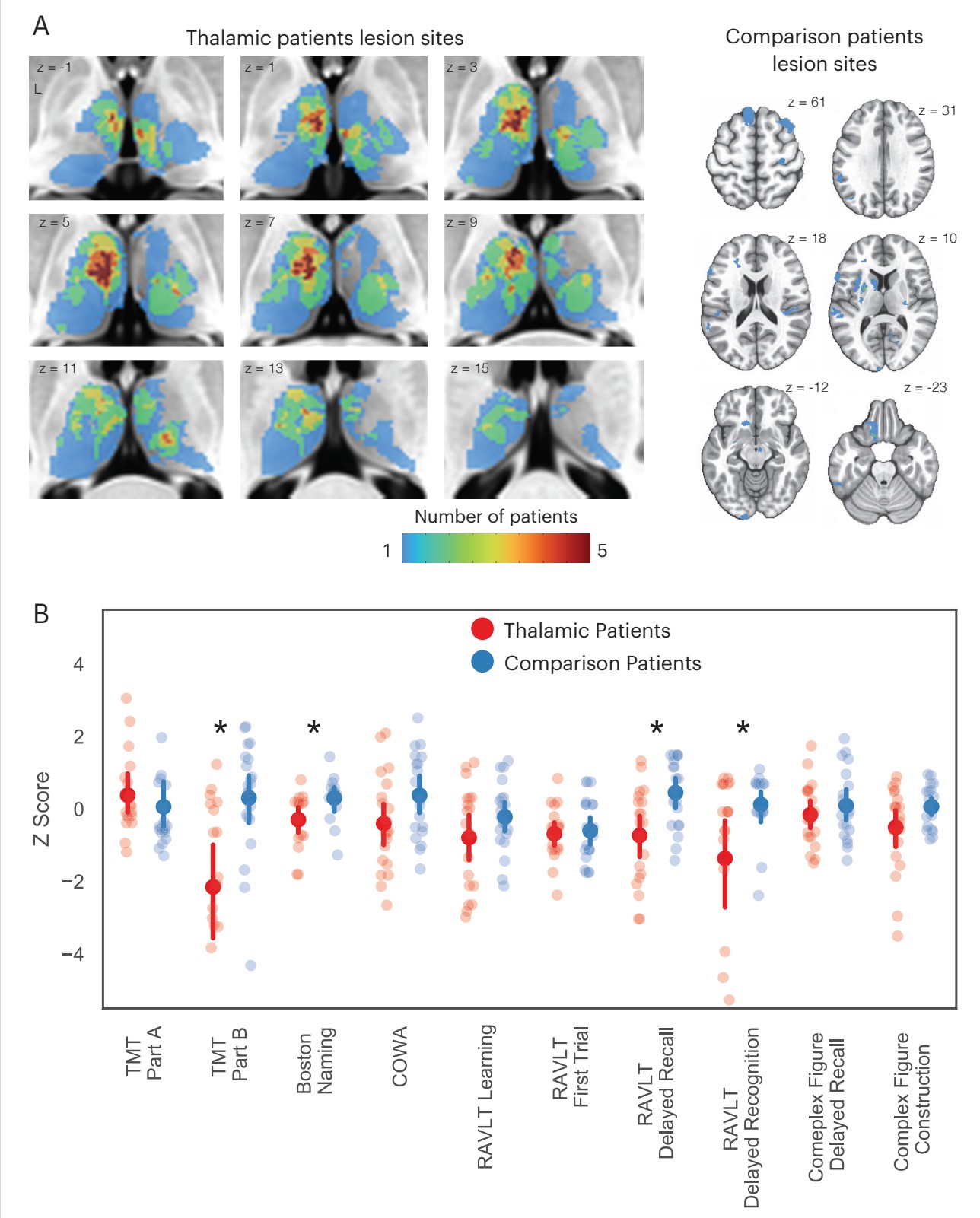

**Figure 1.** Thalamic patients performed significantly worse on multiple neuropsychological tests compared to comparison patients. (**A**) Overlap of lesions in patients with thalamic lesions and comparison patients with cortical lesions. (**B**) Neuropsychological test scores from patients with thalamic lesions and comparison patients. All test scores transformed to z-score using published population norm. We inverted z-scores from TMT Part B to facilitate comparison, for all tests negative z-scores indicate more severe impairment. For each plot, the solid dot depicts the mean, and the bar depicts

*Figure 1 continued on next page*

*Figure 1 continued*

the 95 % bootstrapped confidence interval. * Indicates corrected p < 0.05. TMT: Trail Making Test; COWA: Controlled Oral Word Association Test; RAVLT: Rey Auditory-Verbal Learning Test.

assessing functions from the following domains: (1) executive function using the Trail Making Test Part B (TMT Part B); (2) verbal naming using the Boston Naming Test (BNT); (3) verbal fluency using the Controlled Oral Word Association Test (COWA); (4) immediate learning using the first trial test score from the Rey Auditory-Verbal Learning Test (RAVLT); (5) total learning by summing scores from RAVLT, trials 1–5; (6) long-term memory recall using the RAVLT 30 min delayed recall score; (7) long-term memory recognition using the RAVLT 30 min delayed recognition score; (8) visuospatial memory using the Rey Complex Figure delayed recall score; (9) psychomotor function using the Trail Making Test Part A (TMT Part A); and (10) construction using the Rey Complex Figure copy test. We grouped these tests into the executive (TMT Part B), verbal (COWA and BNT), memory (RAVLT delayed recall and delayed recognition), learning (RAVLT immediate learning and total learning), psychomotor (TMT Part A), and visuospatial (Complex Figure tests) domains (*Lezak et al., 2012*). To adjust for demographic factors such as age and years of education, all test scores were transformed to z-scores using published population norms. Statistical significance of between-group comparisons were assessed with the randomized permutation tests unless otherwise noted.

We found that patients with thalamic lesions performed significantly worse than comparison patients on the following tests (*Table 1*): TMT Part B, BNT, RAVLT delayed recall, and RAVLT delayed recognition. Notably, each test had at least one thalamus patient that performed worse than 95 % of the normative population (z-score <–1.645; *Figure 1B*). These results suggest that thalamic lesions were associated with more severe behavioral impairments in executive, verbal, and memory functions relative to comparison patients that had damage outside of the thalamus.

There are two potential models of thalamocortical connectivity that could explain the observed behavioral impairments. First, each cognitive domain is associated with a distinct thalamocortical system, thus different task impairments are associated with segregated lesion sites within the thalamus. Alternatively, thalamic hubs are involved in many cognitive processes across domains through their widespread connectivity with multiple systems, and thus lesions to a critical hub region could be associated with widespread impairment. The first explanation predicts little to no lesion overlap among different impaired tasks, whereas the second model predicts a high degree of lesion overlap

**Table 1.** Group comparisons on neuropsychological test results.

| | Thalamic patients | | Comparison patients | | |
| --- | --- | --- | --- | --- | --- |
| | Mean z-score | SD | Mean z-score | SD | Randomized permutation p |
| Trail Making Test Part A | 0.38 | 1.08 | 0.06 | 1.48 | 0.10 |
| Trail Making Test Part B | –2.15 | 2.94 | 0.3 | 1.55 | <0.001 |
| Boston Naming Task | –0.29 | 0.73 | 0.3 | 0.64 | 0.0033 |
| Controlled Oral Word Association Test | –0.4 | 1.3 | 0.38 | 1.16 | 0.055 |
| *Rey Auditory-Verbal Learning* | | | | | |
| Immediate Learning | –0.68 | 0.71 | –0.6 | 0.85 | 0.76 |
| Total Learning | –0.79 | 1.37 | –0.22 | 0.92 | 0.11 |
| Delayed Recall | –0.73 | 1.29 | 0.45 | 0.94 | 0.0021 |
| Delayed Recognition | –1.36 | 2.7 | 0.12 | 0.86 | 0.0049 |
| *Rey Complex Figure Test* | | | | | |
| Delayed Recall | –0.16 | 0.86 | 0.094 | 0.94 | 0.41 |
| Copy Test | –0.51 | 1.17 | 0.064 | 0.56 | 0.098 |

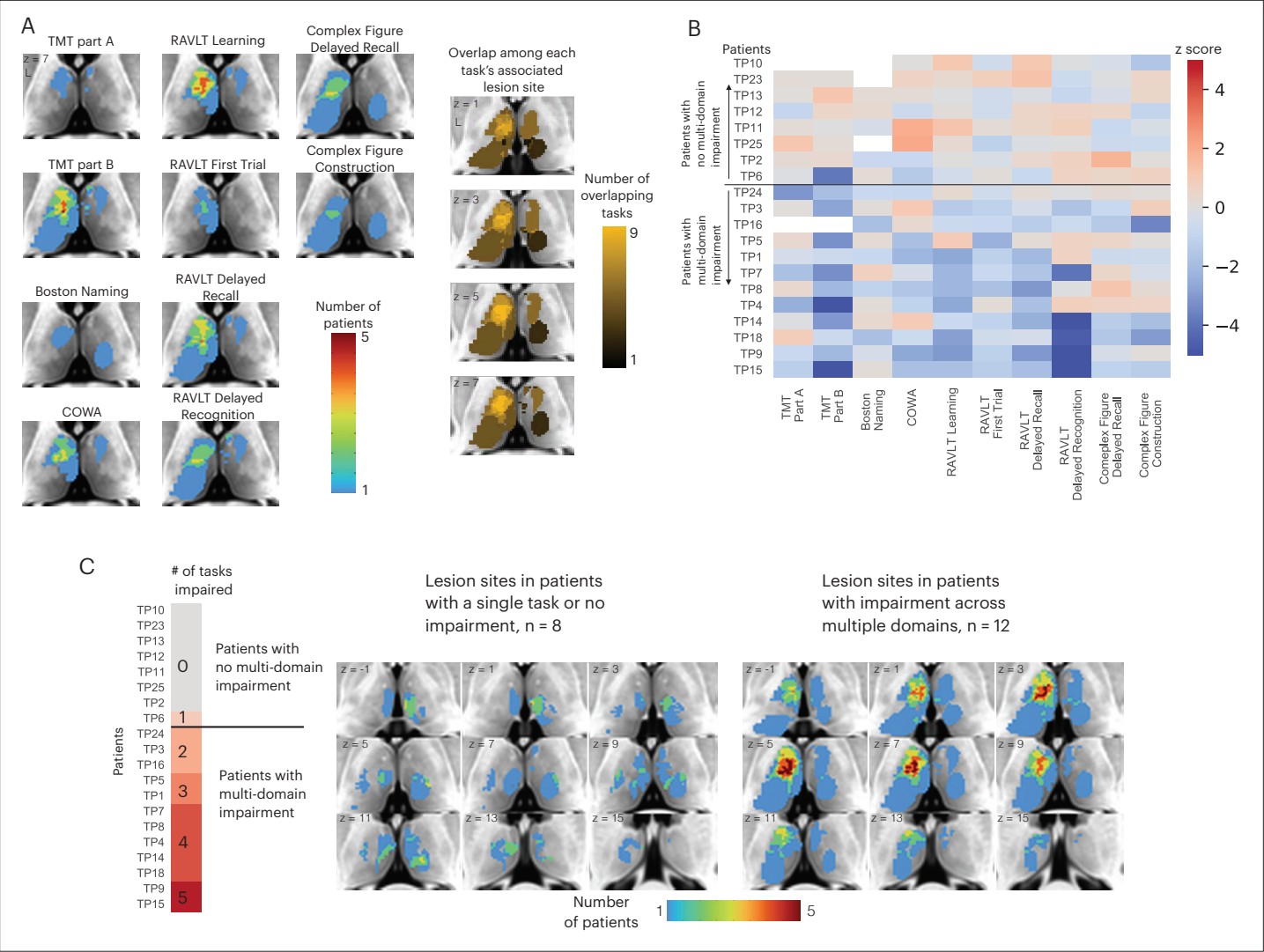

**Figure 2.** Lesions associated with impaired performance on tasks across multiple cognitive domains. (**A**) Left panel: overlap of lesion masks from subjects with impaired test performance on each task. Impaired task performance defined as z < −1.645 (95 percentile in z distribution). Right panel: overlap of lesion sites associated with impairment on each individual task (summing each individual task's lesion map from the left panel). (**B**) Table showing each thalamus patient's task performance on 10 different neuropsychology tests. For all tasks, negative z-scores indicate more pronounced impairment. Both Trail Making Test (TMT) Part A and Part B scores were inverted to match the directionality of other tests. (**C**) Left panel: classifying thalamic patients into groups that exhibit impairment in one versus multiple tasks across cognitive domains. Right panel: Lesion sites in patients with or without impairment across multiple tasks.

The online version of this article includes the following figure supplement(s) for figure 2:

**Figure supplement 1.** Lesion sites of all thalamic patients.

among the impaired tasks. To discern between these two possibilities, we first examined lesion sites associated with impaired performance separately for each task (*Figure 2A*, left panel). Notably, we found an overlapping lesion site in the left anterior-medio-dorsal thalamus that is associated with impairment across different cognitive domains (*Figure 2A*, right panel). This result suggests that a patient with a focal thalamic lesion to this multi-domain lesion site could exhibit behavioral impairment across cognitive domains.

We plotted the degree of impairment (expressed by z-score) across all tasks and cognitive domains separately for each patient (*Figure 2B*), and found that in 12 out of 20 patients, significant impairment (z < −1.645) was reported in more than two cognitive domains. We then examined whether patients with behavioral impairments across multiple domains had lesions to this identified overlapping site (*Figure 2C*; for each patient's lesion, please see *Figure 2—figure supplement 1*). We found that in

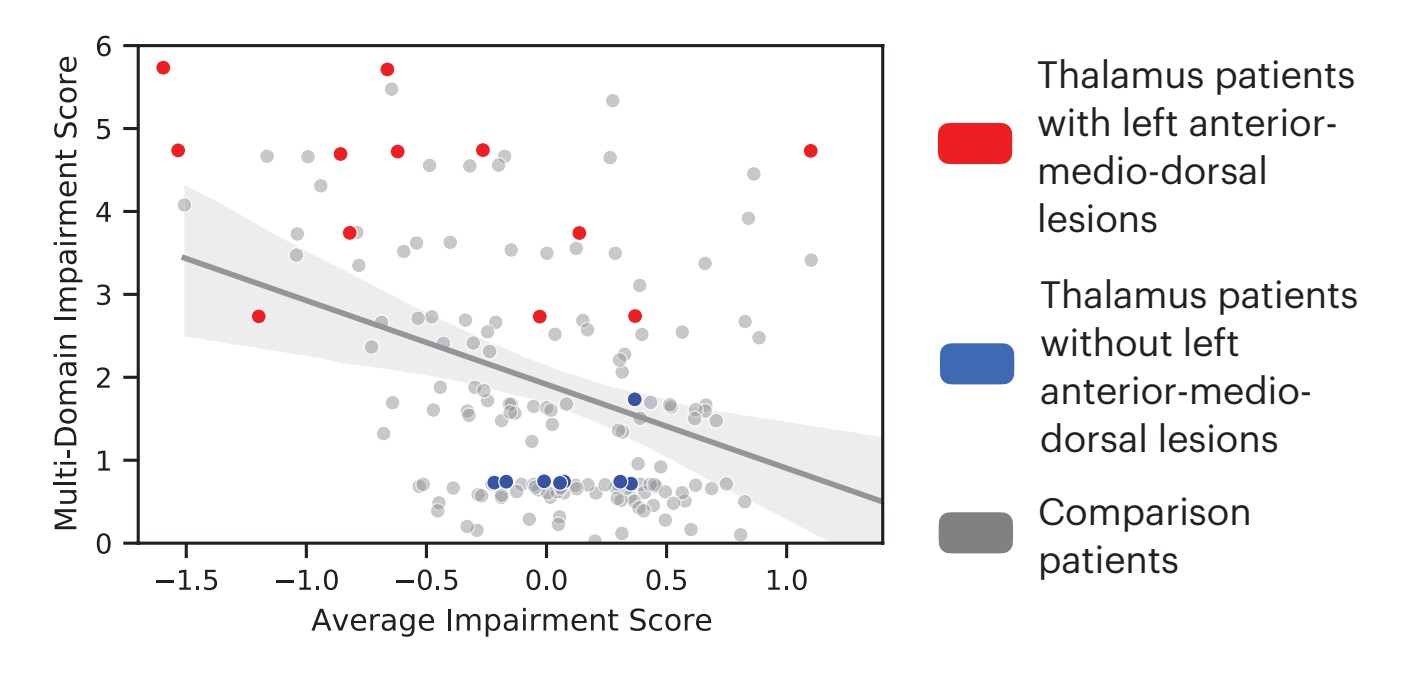

**Figure 3.** Comparing the degree of average behavioral impairment (x axis) and multi-domain behavioral impairment (y axis) between thalamus patients and comparison patients. More negative average impairment score represents more severe behavioral impairment. Higher multi-domain impairment score indicate more cognitive domains were affected. Individual dots represent individual patients. The solid line indicates a fitted regression line for the comparison patient group. The shaded area represents the 95 % confidence interval.

The online version of this article includes the following figure supplement(s) for figure 3:

**Figure supplement 1.** Lesion sites 145 comparison patients.

the 12 patients that exhibited impairment across multiple domains, there was indeed an overlapping lesion site in the left anterior-medio-dorsal thalamus. This overlapping site was notably absent in the eight patients that exhibited either no behavioral impairment or impairment in one single task (*Figure 2C*, left panel). This pattern was also observed when examining lesion sites from individual patients, as individual patients with impairments across multiple domains had lesions that overlapped with this left anterior-medio-dorsal thalamic region (Montreal Neurological Institute [MNI] coordinate: –7, 10, 8).

We further tested whether lesions to the left anterior-medio-dorsal thalamus were associated with broad impairment across more cognitive domains rather than being driven by more severe deficits in a limited number of domains. For this purpose, we included an expanded group of 145 comparison patients from the Iowa Neurological Patients Registry (ages 19–81 years, 67 males; for lesion coverage, see *Figure 3—figure supplement 1*). Unlike the first group of comparison patients (*Figure 1*), these comparison patients were not matched with the thalamus patients on the lesion size, but on the averaged severity of behavioral deficits across all 10 neuropsychology tests. We predicted that when matched on the severity of behavioral deficit, patients with lesions that over-lapped with the left anterior-medio-dorsal thalamus would exhibit impairments on more cognitive domains, whereas comparison patients will exhibit more circumscribed deficits in fewer cognitive domains. To test this prediction, we first calculated an 'average impairment score' by averaging the normalized z-scores across all 10 neuropsychological tests, and a 'multi-domain impairment score' by summing the number of tests with significant behavioral deficits (defined as $z < 1.645$). Both scores regressed out the variance associated with differences in lesion size. We then fitted a linear regression model to the group of 145 comparison patients that had average impairment scores similar to thalamus patients (maximum = 1.39, minimum = –1.59). We found that 11 out of 12 thalamus patients with lesions that overlapped with the left anterior-medio-dorsal thalamus (*Figure 2C*) exhibited higher multi-domain impairment score when compared to comparison patients (*Figure 3*). This suggests that lesions to the left anterior-medio-dorsal thalamus are not merely associated with more severe

behavioral impairment, but also associated with impairment across multiple cognitive domains to a greater extent than would be expected from lesions in other brain regions.

Because it is possible that impairment across cognitive domains was driven by larger lesions that damaged many functionally specialized subregions in the thalamus, we tested whether there was a significant association between lesion size and the extent of behavioral impairment. We found that there was no significant correlation between lesion volume and number of cognitive domains impaired (r(19) = 0.21, p = 0.36). Furthermore, patients with impairment in more than two cognitive domains did not have larger lesions when compared to patients with impairment in less than two cognitive domains (randomized permutation test p = 0.42). Using the Morel thalamic nuclei atlas, we compared the number of anatomical nuclei in the thalamus overlapped with lesions associated with and without multi-domain impairment. Thalamic nuclei were defined using a published atlas derived from postmortem human brains (*Krauth et al., 2010*). We found no significant difference (multi-domain lesions: mean = 4.9 nuclei, SD = 1.89; single-domain lesions: mean = 5.92 nuclei, SD = 2.7; randomized permutation p = 0.39). Using a thalamic functional parcellation atlas we previously published that segmented the thalamus into different network parcellations (*Hwang et al., 2017*), we also did not find significant differences in the number of parcellations between these lesion sites (multi-domain lesions: mean = 3.25 parcels, SD = 1.87; single-domain lesions: mean = 3.88 parcels, SD = 2.59; randomized permutation p = 0.21; randomized permutation p = 0.62). These results indicate that thalamic lesions associated with more global deficits did not involve more thalamic subregions than lesions that did not.

We further evaluated the brain activity maps likely associated with the putative cognitive processes assessed by each of the neuropsychology tests we assessed. Specifically, we utilized the Neurosynth database (*Yarkoni et al., 2011*), which contains activation loci from thousands of published fMRI studies, to perform automatic meta-analyses and identify brain regions likely recruited for each task. We queried the following terms: 'executive function', 'recall', 'recognition', 'fluency', 'naming'. We found that these terms were associated with distinct brain activity maps with minimal overlap (*Figure 4*). Maps for terms 'naming' and 'fluency' overlapped in the left frontal cortex, maps for 'recall' and 'recognition' overlapped in temporal cortices, and all four maps overlapped in a small region (three 2 mm$^3$ voxels) in the left inferior frontal gyrus.

We performed additional analyses to contrast the hub properties between thalamic lesion sites that were only observed in patients that exhibited impairment across multiple domains, versus lesion sites only observed in patients that exhibited no behavioral impairment or impairment in a single domain (*Figure 5A*). Lesion sites that overlapped between patients that exhibited multi-domain impairment and patients with no behavioral impairment were excluded. Our prediction was that lesions to thalamic regions exhibiting prominent hub properties—as measured with PC—will have an association with widespread negative behavioral outcomes across multiple domains. In contrast, lesions to thalamic regions with lower hub properties will have less widespread association. We calculated the PC value for each thalamic voxel using a large normative thalamocortical functional connectome dataset (*Hwang et al., 2017*). The purpose was to estimate hub properties of thalamic subregions in the healthy population and use this result to estimate hub properties of the lesion sites. Briefly, for each normative subject, we calculated thalamocortical functional connectivity between thalamic voxels and 400 cortical regions of interests (ROIs), spanning seven canonical cortical networks (*Schaefer et al., 2018*). We did not calculate functional connectivity between thalamic voxels. Voxel-wise PC values were calculated for each subject and averaged across subjects. A high PC value indicates that a hub region has distributed connectivity with multiple cortical networks. We found that when averaged across normative subjects, the anterior, medial, and dorsal thalamus exhibit strong connector hub properties (*Figure 5B*). We then compared the distribution of voxel-wise PC values between the multi-domain and single-domain lesion sites using the Komogorov-Smirnov test, and found that thalamic voxels in the multi-domain lesion sites had on average higher PC values compared to those in the single domain lesion sites (*Figure 5C*). The voxel-wise PC values were significantly higher in the multi-domain sites compared to the single-domain sites (Kolmogorov-Smirnov d = 0.16, p < 0.001; replication dataset: Kolmogorov-Smirnov d = 0.11, p < 0.001). We then statistically compared PC values between multi-domain and single-domain sites across individual subjects from the normative cohort, and found that the multi-domain sites exhibited significantly higher PC values (*Figure 5D*, t(234) = 6.472, p < 0.001; replication dataset: t(61) = 3.21, p = 0.002), confirming our central prediction.

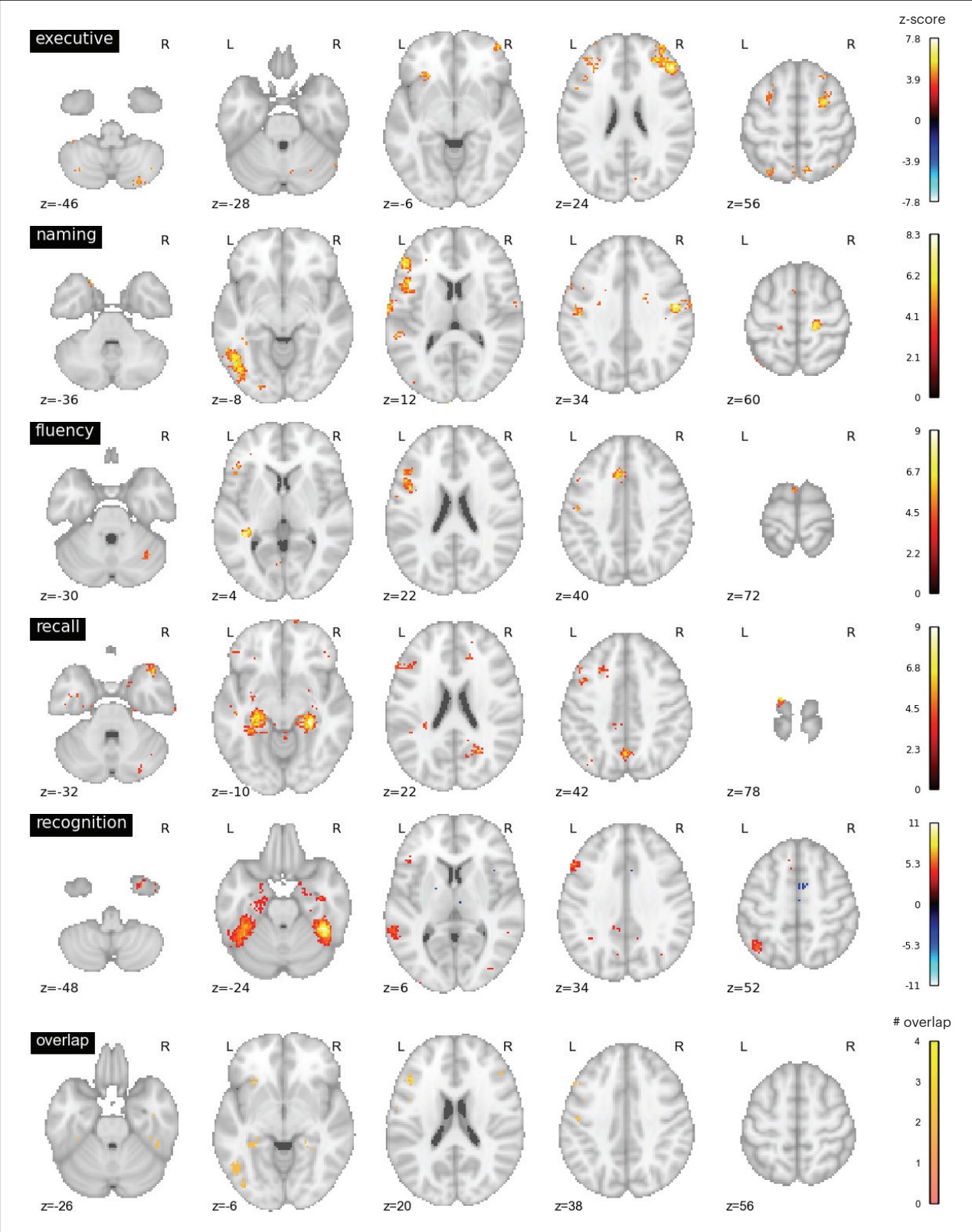

**Figure 4.** Neurosynth meta-analyses. Top four rows: brain regions associated with putative cognitive processes assessed by the Trail Making Test Part B (TMT Part B), Boston Naming Test (BNT), Controlled Oral Word Association Test (COWA), Rey Auditory-Verbal Learning Test (RAVLT) delayed recall, and RAVLT delayed recognition tests. Color bar represents the strength of association in z-score. Bottom row: overlap between the top four maps. Maps for 'naming' and 'fluency' overlapped in the left frontal cortex, maps for 'recall' and 'recognition' overlapped in temporal cortices. All four maps overlapped with six voxels in the left middle frontal gyrus.

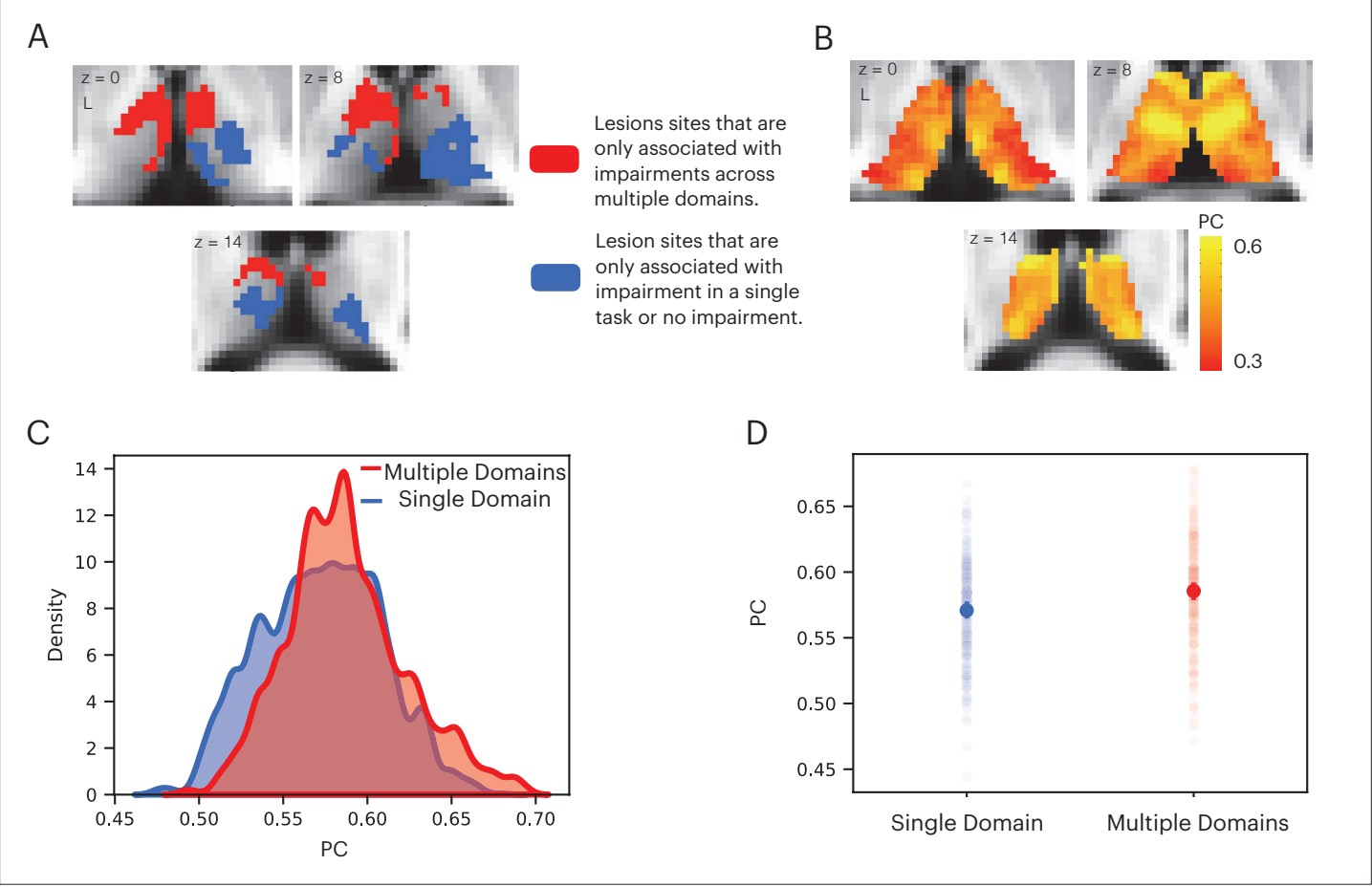

**Figure 5.** Lesions to thalamic regions with strong hub properties are associated with behavioral impairment across cognitive domains. (**A**) Thalamic lesion sites associated with multi-domain impairment are located in the anterior-medio-dorsal thalamus. (**B**) Right panel: left medial and anterior thalamus is associated with prominent hub property (measured by participation coefficient [PC]). (**C**) Kernel density plot of voxel-wise PC values from thalamic lesion sites associated with multi-domain and single-domain impairment. Voxel-wise PC values were significantly higher for multi-domain lesion sites. The y-axis was scaled so area under the curve is summated to 1. (**D**) In a group of 235 subjects, PC values were significantly higher in multi-domain versus single-domain lesion sites in the thalamus. Each data dot indicates PC value from one normative subject.

Lesions to cortical hubs are known to be associated with widespread behavioral impairments across multiple cognitive domains (**Warren et al., 2014**; **Reber et al., 2021**). To replicate this finding and validate our approach with the larger expanded comparison patient group (same sample as **Figure 3**), we examined whether comparison lesions associated with multi-domain behavioral impairments also exhibited stronger connector hub properties (higher PC values). We grouped the 145 expanded comparison patients into two sub-groups: those with (n = 58) and without (n = 87) behavioral impairments in more than one cognitive domain. We then mapped the lesion sites associated with these two patient groups, and found that comparison lesions associated with multi-domain impairment were primarily located in lateral frontal and posterior parietal associative regions, regions prior studies found to exhibit strong connector hub properties (**Figure 6A**). To assess the hub property of each comparison patient's lesion, we calculated the PC value of each cortical GM voxel from the normative functional connectome (**Reber et al., 2021**). Consistent with findings from prior studies (**Bertolero et al., 2015**; **Power et al., 2013**), the lateral frontal and posterior parietal regions exhibit strong connector hub properties (**Figure 6B**). We then compared the distribution of voxel-wise PC values between the multi-domain and single-domain lesion sites in comparison patients, and found that voxels in the multi-domain lesion sites had on average higher PC values compared to those in the single-domain lesion sites (**Figure 6C**; Kolmogorov-Smirnov d = 0.11, p < 0.001; replication dataset: Kolmogorov-Smirnov d = 0.14, p < 0.001).

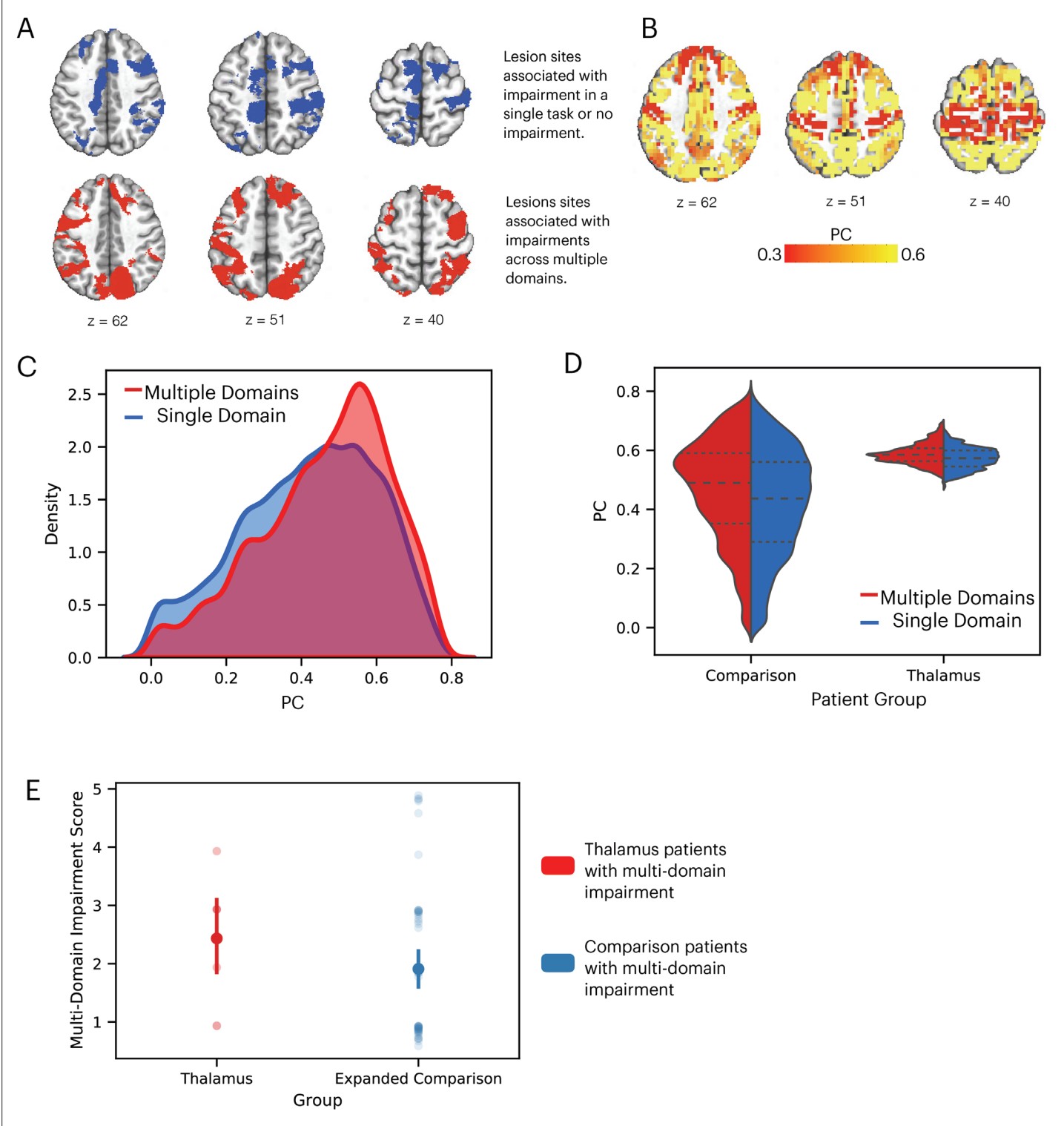

**Figure 6.** Comparison lesions with strong hub properties are associated with behavioral impairment across cognitive domains. (**A**) Comparison lesion sites associated with multi-domain impairment are located in the lateral frontal and posterior parietal regions. (**B**) Frontal and parietal regions exhibit strong connector hub properties (measured by participation coefficient [PC]). (**C**) Kernel density plot of voxel-wise PC values from comparison lesion sites associated with multi-domain and single-domain impairment. Voxel-wise PC values were significantly higher for multi-domain lesion sites. The y-axis was scaled so area under the curve is summated to one. (**D**) Comparing the distribution of PC values in comparison and thalamic lesion sites. (**E**) Comparing multiple-domain impairment score between thalamic and expanded comparison patients with multi-domain impairment.

We then compared the hub properties between thalamic lesions and comparison lesions associated with multi-domain impairment (*Figure 6D*), and found the comparison lesions had a wider distribution of PC values, likely because comparison lesions included in *Figure 6* were significantly larger than thalamic lesions (comparison lesions: mean = 36440 mm$^3$, SD = 33078 mm$^3$, thalamic lesions: mean = 1559 mm$^3$, SD = 1312 mm$^3$, randomized permutation test p < 0.001). The peak values were comparable between thalamic lesions and comparison lesions associated with multi-domain impairment (PC = 0.57). We found no statistically significant difference in the multiple-domain impairment score between thalamic and expanded comparison patients with multi-domain impairment (*Figure 6E*, randomized permutation test p = 0.17).

Findings reported in *Figure 5* suggest that lesion sites associated with impairment across cognitive domains have a diverse functional relationship with distributed systems involved in different cognitive functions. Thus, we mapped the cortical functional networks that show strong functional connectivity with voxels within the multi-domain lesion site. For every thalamic voxel, we calculated its average functional connectivity with seven cortical functional networks (*Schaefer et al., 2018*), including the visual, somatomotor (SM), limbic, dorsal attention (DA), cingulo-opercular (CO), FP, and default mode (DMN) networks. We then divided the functional connectivity estimates of each network by the total summated functional connectivity strength of each voxel. The purpose of this procedure was to derive a functional connectivity weight ratio estimate to assess the network selectivity of each voxel. If a voxel only interacts with a specific network, the majority of its functional connectivity strength should be devoted to that network, resulting in a high functional connectivity weight ratio, whereas connectivity with other networks should be considerably lower. In contrast, if a voxel is broadly interacting with multiple functional networks, then it should exhibit overlapping functional connectivity weight ratios for those networks. Consistent with the high PC values we observed, we found that thalamic voxels in the multi-domain lesion sites exhibit a diffuse functional connectivity relationship with cortical functional networks that are predominately located in heteromodal association areas, including the FP, DMN, limbic, and CO networks (*Figure 7A*). The centroids of functional connectivity weights were between 0.15 and 0.3 for these networks, and lower (close to 0) for the visual, SM, and DA networks.

We repeated the functional connectivity weight ratio analysis to assess functional connectivity between thalamic voxels and cortical regions identified via the Neurosynth meta-analyses (*Figure 4*). Specifically, we calculated the functional connectivity between each thalamic voxel and cortical voxels associated with each queried term ('executive', 'naming', 'fluency', 'recall', 'recognition'), then divided the averaged functional connectivity estimates of each term by the total summated functional connectivity strength of each voxel. The purpose was to determine whether there are voxels only selectively interacting with brain regions associated with specific cognitive processes. We found a distributed thalamocortical functional connectivity relationship with brain regions associated with these putative cognitive processes (*Figure 7B*). Higher functional connectivity weight ratios were found for regions implicated in 'executive' and 'fluency', and lower ratio for 'naming', 'recall', and 'recognition'. Notably, we did not observe a pattern that would indicate strong functional specificity, as the highest observed weight ratio was less than 37% . This result suggests that the thalamic hub region is not selectively interacting with a specific cortical system, instead it diffusely interacts with distinct brain regions associated with these varied cognitive processes.

The thalamus is also known to have dozens of constituent nuclei that can be defined using chemo-architectual and cytoarchitectual properties. We further examined which thalamic nuclei overlapped with the multi-domain lesion sites, by calculating the percentage of lesioned voxels within the multi-domain lesion sites for each thalamic nucleus (*Krauth et al., 2010*). We found that the multi-domain lesion sites overlap with several higher-order thalamic nuclei, including the anterior nucleus (AN), the MD, the ventromedial nucleus (VN), the intralaminar nucleus (IL), as well as other nuclei, including the ventro-anterior (VA) and the ventrolateral (VL) nuclei (*Figure 7C*).

Thalamic subdivisions are comprised of a mixture of 'core' and 'matrix' thalamocortical projections cells, and each type has a different projection pattern to the cortex (*Jones, 2009*; *Jones, 2001*). Importantly, these cell types also express distinct calcium-binding proteins: Calbindin-rich matrix cells project diffusely to the superficial layers of the cerebral cortex, unconstrained by functional borders between cortical regions; whereas in contrast, parvalbumin-rich core cells send topographic specific projections to the middle layers of the cerebral cortex. The distributed projection pattern of matrix cells is conceptually analogous to the inter-network connectivity property of connector hubs. Therefore,

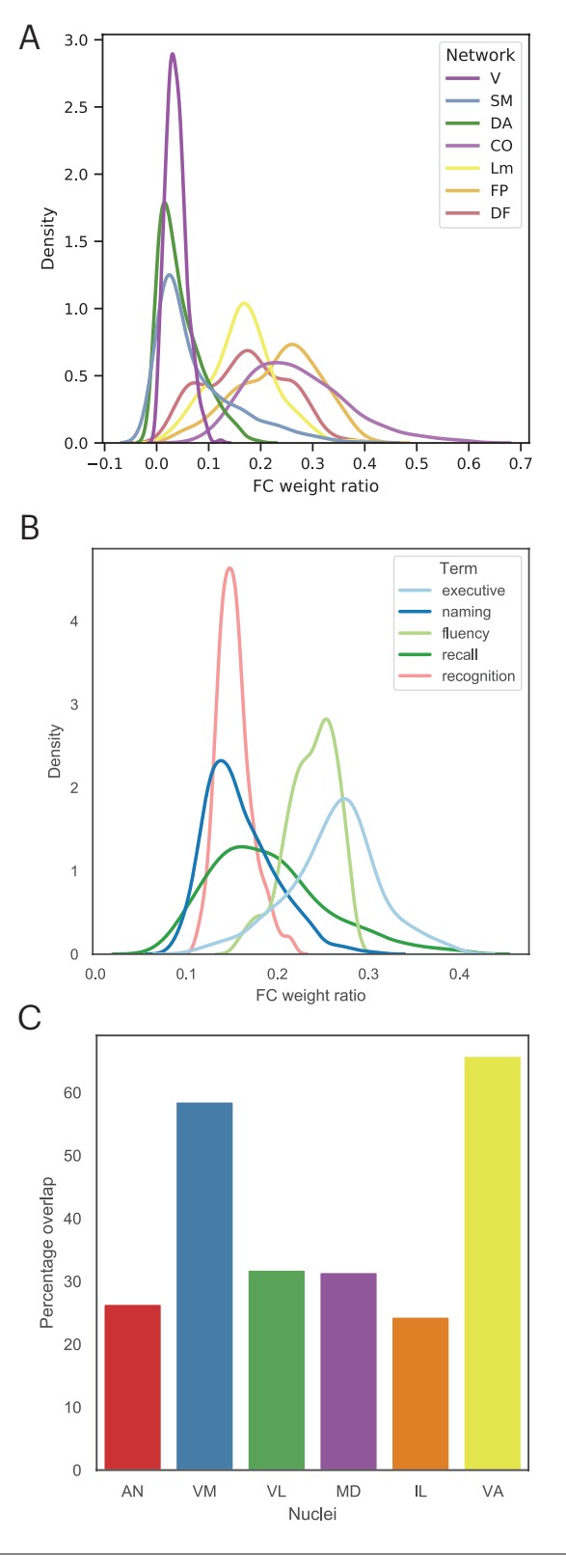

**Figure 7.** The multi-domain lesion site does not show strong functional specificity. (**A**) Voxel-wise kernel density plot of thalamocortical functional connectivity weight for each cortical network. Voxels in the multi-domain lesion site exhibit stronger functional connectivity with the cingulo-opercular (CO), limbic (Lm), frontoparietal (FP), and default mode (DMN) networks. Weaker connectivity with visual (**V**), somatomotor (**SM**), and dorsal attention (DA)

*Figure 7 continued*

networks. (**B**) Voxel-wise kernel density plot of thalamocortical functional connectivity weight with cortical regions identified via Neurosynth meta-analyses. For both (**A**) and (**B**), higher weight ratio indicates functional specificity, suggesting that voxel is selectively interacting with a specific cortical system. (**C**) The multi-domain lesion site overlaps with higher-order thalamic nuclei, including the anterior (AN), ventromedial (VM), mediodorsal (MD), intra-laminar (IL), ventro-anterior (VA), and ventrolateral (VL) nuclei.

we further predicted that the multi-domain lesion sites would contain relatively higher concentrations of matrix cells when compared to the single-domain lesion sites. To test this prediction, we examined data from the Allen Human Brain Atlas that estimated the brain-wide expression of calbindin and parvalbumin proteins, CALB1 and PVALB, respectively (*Gryglewski et al., 2018*). This allowed us to estimate the relative density of matrix and core cells in each thalamic voxel (*Müller et al., 2020*). We found that CALB1 were more strongly expressed in the anterior-medio-dorsal thalamus (*Figure 8A*), suggesting that multi-domain lesion sites likely contain higher densities of matrix projection cells. We then compared the distribution of voxel-wise CALB1 expression between the multi-domain and

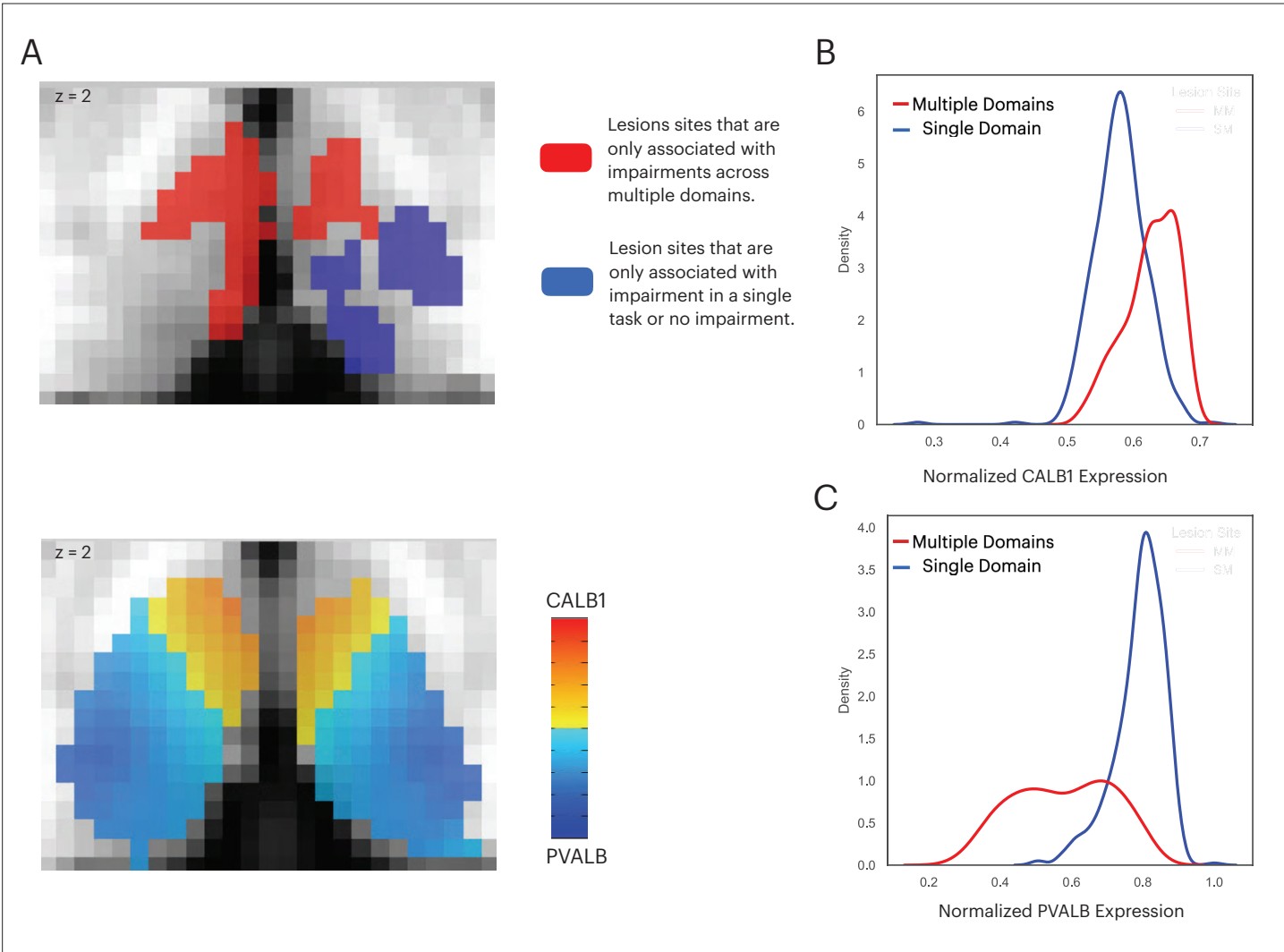

**Figure 8.** The multi-domain lesion site contains higher concentration of matrix cells. (**A**) The anterior-medial thalamus had a relative higher expression of CALB1 (lower panel), which overlapped with the multi-domain lesion sites (upper panel). Color intensity in the lower panel denotes the relative expression of CALB1 and PVALB in different thalamic voxels (*Jones, 2009*; *Jones, 2001*). (**B**) Voxel-wise kernel density plot of normalized CALB1 expression level for the multi-domain versus the single-domain lesion sites. (**C**) Voxel-wise kernel density plot of normalized PVALB expression level for the multi-domain versus the single-domain lesion sites.

single-domain lesion sites using the Komogorov-Smirnov test, and found that thalamic voxels in the multi-domain lesion sites had on average higher CALB1 expression when compared to those in the single-domain lesion sites (*Figure 8B*; Kolmogorov-Smirnov d = 0.483, p < 0.001). This suggests that the multi-domain lesion sites had more matrix cells when compared to the single-domain lesion sites. In contrast, thalamic voxels in the single-domain lesion sites had on average higher PVALB expression compared to those in the multi-domain lesion sites (*Figure 8C*; Kolmogorov-Smirnov d = 0.482, p < 0.001), suggesting relatively higher concentrations of core cells in the single-domain lesion sites. We then calculated the Dice coefficient to compare the spatial overlap between the CALB1 expression map and lesion masks. We found the CALB1 expression map to be more similar to the multi-domain lesion mask (Dice coefficient = 0.49), and less similar to the single-domain lesion site (Dice coefficient = 0.007).

## Discussion

Prior studies that examined the network organization of the human brain have consistently identified the thalamus as a prominent hub. Within the thalamus there are regional differences in relative hubness, with the anterior, medial, and dorsal thalamus found to exhibit the strongest hub property and broadly connected with distributed functional networks (*Cole et al., 2010*; *Hwang et al., 2017*). This connectivity architecture likely allows the thalamus to flexibly participate in functions that support cognition across multiple domains. Findings from the current study provide empirical evidence that confirms the behavioral significance of this network architecture.

The lesion method is well suited for determining the relationship between thalamic hubs and cognition. Lesioning a hub region should weaken network interactions across multiple systems and have a widespread influence on behavior across cognitive domains. In contrast, lesioning a thalamic subregion with a specific cognitive function should have a more limited effect. Past studies have found that lesions to the human thalamus are associated with a wide range of cognitive impairments, including executive dysfunction (*Hwang et al., 2020*; *Liebermann et al., 2013*), amnesia (*Graff-Radford et al., 1990*; *Pergola et al., 2016*; *von Cramon et al., 1985*), aphasia (*Crosson et al., 1986*; *Graff-Radford et al., 1984*), and attention deficits (*de Bourbon-Teles et al., 2014*; *Snow et al., 2009*). However, it is not clear whether deficits reported in prior studies each localize to distinct thalamic regions, or whether a restricted lesion to a hub region can be associated with widespread effects. Our findings help to answer this question.

Specifically, we found significant impairment on the TMT Part B, RAVLT, and BNT tests in patients with lesions to an overlapping region in the left anterior-medio-dorsal thalamus. Of note, basic perceptual and motor functions in thalamic patients were comparable to comparison patients, given that we did not find a significant difference in the TMT Part A and Rey Complex Figure construction test. Thus, the impairment we observed is related to higher-order cognitive processes related to language, memory, and executive functions. How can a thalamic subregion be simultaneously associated with executive, language, and memory functions? Our framework suggests that each of these cognitive domains is associated with a different brain system, and these brain systems have converging connectivity with the multi-domain lesion site that we identified. In support, we found that this lesion site exhibited a strong connector hub property, with a diverse functional connectivity relationship with multiple cortical networks. Importantly, the connector hub property of this multi-domain lesion site was greater than lesion sites that were associated with more limited impairment, again confirming the behavioral relevance of network hubs.

We suspect that the left thalamus is involved because language-related functions are more lateralized to the left hemisphere, and cognitive performance using standard neuropsychological tests tends to be left-lateralized (*Adolphs et al., 2020*). Successful performance on the delayed recall and delayed recognition tests likely requires a long-term memory system within which the anterior thalamus is a pivotal component, interlinking the hippocampal and cortical systems (*Aggleton et al., 2010*). Prior studies on lesion symptom mapping have found that TMT and language-related tasks are associated with medial frontal, lateral frontal, and temporal cortices (*Baldo et al., 2013*; *Glascher et al., 2012*). These brain regions likely overlap with the FP, CO, and DF networks that we found to have strong functional connectivity with the multi-domain lesion site. Furthermore, these cortical regions are known to receive anatomical projections from the AN and MD, which we also found to overlap with the identified lesion site (*Barbas et al., 1991*; *Giguere and Goldman-Rakic, 1988*; *Selemon and*

*Goldman-Rakic, 1988*). In addition to the AN and MD, the multi-domain lesion sites also overlapped with the IL and VN. These thalamic nuclei are known to have higher densities of matrix projection cells (*Jones, 2009*) that express Calbindin-binding proteins. Matrix cells broadly project to multiple cortical regions crossing receptive fields and functional boundaries (*Jones, 2001*). This indicates that matrix cells simultaneously project to segregated and distributed cortical regions, an anatomical feature that is consistent with the properties of brain network hubs (*Müller et al., 2020*). Therefore, matrix cells may be the anatomical connectivity substrate that allows the anterior-medio-dorsal thalamus to promote its connector hub functions that we will discuss below.

How do thalamic connector hubs support cognitive functions across these diverse domains? Studies on brain network organization may offer some insights. For example, brain functions engage functional segregation, where segregated systems can perform specialized functions without interference, and functional integration, where outputs from specialized systems, can be integrated between domains via connector hubs that interlink distributed systems (*Bertolero et al., 2015*; *Cohen and D'Esposito, 2016*; *Shine et al., 2016*). This modular-like organization balances segregated and integrated processes and can be reliably observed in the human brain (*Meunier et al., 2010*; *Sporns and Betzel, 2016*). Furthermore, an optimal modular structure has been found to correlate with behavioral performance on tasks across multiple domains (*Bertolero et al., 2018*), and lesions to connector hubs, including the thalamus, have been shown to disrupt modular organization more so than other non-hub lesions (*Gratton et al., 2012*; *Hwang et al., 2017*). Therefore, lesions to thalamic hubs may disrupt the optimal organization of multiple systems via its widespread thalamocortical connectivity, and affect both functional segregation and integration. This disruption will not be limited to one system nor to a single cognitive domain, a prediction that is consistent with our finding. Thus, one hypothetical function of thalamic connector hubs is to maintain a cognitively optimal architecture across multiple functional systems to support diverse cognitive processes.

Our results further indicate that we cannot fully dissociate the averaged severity of behavioral impairment from the diversity of behavioral impairment, as patients with multi-domain impairment were also more severely impaired across multiple tests. This suggests that there could be common latent processes that are necessary for performing different neuropsychological tests, and disruptions in these common processes might globally affect behavior without specificity. For example, it is possible that thalamic lesions are associated with distal disruptions ('diaschisis'; *Sutterer and Tranel, 2017*; *Von Monakow, 1911*) of cortical systems involved in general task-control functions that impact behavior across multiple domains. We found that the multi-domain lesion site has strong functional connectivity with FP and CO networks, which are cortical networks hypothesized to be involved in domain general processes such as maintaining task-relevant information, guiding relevant sensory inputs toward the correct behavioral output, and adjusting behavior when errors are made (*Dosenbach et al., 2008*; *Gratton et al., 2018*). It is also possible that thalamic lesions disrupt processing in a hypothetical supraordinate, domain-general cortical system, known as the 'multi-demand system' (*Cole et al., 2013*; *Duncan, 2010*). The multi-demand system consists of a set of medial frontal, lateral frontal, and posterior parietal regions do not have clear functional specializations, but are instead broadly tuned to implement operations that might be required to perform the different neuropsychological tests that we utilized. This system partially overlaps with the CO and FP networks (*Gratton et al., 2018*), and a prior study found strong functional connectivity between this multi-domain system with the anterior and dorsal thalamus (*Assem et al., 2020*), which overlaps with the multi-domain lesion site that we identified. Thus, lesions to thalamic hubs may disrupt functions associated with the CO and FP networks, or the multi-demand system, which in turn globally affect functions that are necessary for cognition across domains.

Evidence for animal models further suggest that thalamic hubs can influence cortical activity. For example, inhibiting the VL or MD in rodents diminished cortical evoked activity in rodents (*Bolkan et al., 2017*; *Guo et al., 2017*), and stimulating the MD can enhance the information coding in rodent's medial frontal cortex. In addition to modulating evoked activity, thalamus may also facilitate inter-regional communication via its diverging hub-like connectivity. For example, it has been found that inactivation of the thalamus diminishes the effectiveness of cortico-cortical communication (*Theyel et al., 2010*; *Zhou et al., 2016*). We hypothesize that thalamic modulations of evoked responses, promoting inter-regional communication, maintaining modular organization of functional systems, interacting with task-control networks, may together be generalized processes that are necessary for

a wide range of behavioral tasks. Given that these mechanisms are not domain-specific, damage to thalamic hubs may broadly affect tasks that engage these processes for optimal performance.

One limitation of our study is that we did not have equal sampling of lesion locations throughout different thalamic subregions. Our results could be potentially biased by the higher concentration of anterior and medial thalamic lesions in our sample, and missed other multi-domain hubs (e.g., the posterior-medial thalamus) that we were under-powered to detect. Given that the posterior-medial thalamus has also been found to exhibit strong hub properties (*Hwang et al., 2017*; *Greene et al., 2020*), focal lesions to the posterior-medial thalamus may also be associated with behavioral deficits across multiple behavioral domains. Alternatively, because we found that the anterior but not the posterior thalamus had denser expression of the CALB1 protein, a marker for matrix thalamocortical projection cells, it is also possible that only the anterior-medial but not the posterior-medial thalamus is a multi-domain hub. A negative finding would suggest that PC, a graph metric we and many others used for mapping hubs in the human brain, may need to be combined with sources of data (e.g., behavior and gene expression) to more reliably identify multi-domain hubs. Our study does not have enough data to sufficiently adjudicate these two competing predictions, and this question will have to be addressed by future studies.

It is also possible that each of these neuropsychological tasks recruits a distinct, parallel thalamocortical circuit (*Alexander et al., 1986*), and that larger lesions can cover many small specialized subregions in the thalamus, thus affecting behavior across cognitive domains. Results from ours and previous studies do not support this explanation. First, lesion size did not correlate with the degree of multi-domain impairment. Second, we observed multi-domain impairment in several patients with restricted lesions to the left anterior-medio-dorsal thalamus. Third, the anterior-medio-dorsal lesion site did not cover more thalamic sub-parcellations. Fourth, previous large-scale meta-analyses of fMRI research have found that multiple tasks increased BOLD activity in overlapping thalamic subregions, but not segregated thalamic subregions (*Hwang et al., 2017*; *Yeo et al., 2015*). Fifth, our functional connectivity weight ratio analysis also showed that most, if not all voxels in the multi-domain lesion site have a broad functional connectivity relationship with multiple cortical networks. Finally, Neurosynth queries showed that cognitive processes putatively involved in the impaired neuropsychological tests each associated different sets of spatially segregated brain regions. In other words, the multi-domain lesion site does not appear to have strong functional specificity. Thus, while it is possible that the lesion method does not have the resolution and anatomical specificity to detect lesions that cover many small, spatially restricted, functionally-specific thalamic subregions, our results do not support this interpretation. Instead, the cross-domain impairment that we observed were likely associated with lesions to thalamic hubs that have a converging relationship with multiple systems.

To conclude, the principal contribution of our study is to demonstrate the behavioral significance of thalamic hubs. We found that a thalamic hub in the left anterior-medio-dorsal thalamus is critical for memory, executive, and language-related functions. This significant behavioral profile supports the prominent network position of the thalamic hub. These findings lead to the question: what is the function that is implemented by a thalamic hub that allows it to be broadly involved in cognition? Our findings constrain the possible answers—it must be processes that are not specific to a single cognitive function but can be generalized across domains. As discussed above, one possibility is that thalamic hubs maintain an optimal functional network structure, to promote both segregated and integrated functions, which are domain-general network processes that support cognition across multiple domains.

## Materials and methods
### Subjects

We studied 186 neurological patients (mean age = 51.59, age range = 18–81 years, SD = 13.65 years, 97 males). These participants were selected from the Iowa Neurological Patient Registry, and had focal lesions caused by ischemic stroke, hemorrhagic stroke, or benign tumor resection. Patients with learning disabilities, substance abuse, or premorbid personality disorders were excluded from the study. Neuropsychological assessment was conducted at least 3 months post-lesion onset. We first identified 20 patients with lesions restricted in the thalamus caused by ischemic or hemorrhagic stroke (age = 18–70 years, mean = 55.8 years, SD = 13.94 years, 13 males). In addition to patients with focal

lesions within the thalamus, we included comparison patients that had lesions outside of the thalamus, in an attempt to control for lesion effects not specific to the thalamus. Given that lesion size correlates with behavioral impairment (*Reber et al., 2021*), we first tried to minimize any bias that could be introduced by different lesion sizes between the two patient groups. Thus, the first group of 21 comparison patients had lesion sizes that were equal to or smaller than the largest lesion size we observed in the thalamus group (ages 19–77 years, mean = 52.66 years, SD = 11.51 years, 8 males). We further included an expanded group of comparison patients with 145 patients (ages 19–81, mean = 50.83 years, SD = 13.98 years, 67 males). The expanded comparison patients had lesions outside the thalamus, not matched in lesion size, but on the average overall severity of behavioral deficit relative to the thalamus patients. Demographic data for all patients are presented in . All participants gave written informed consent, and the study was approved by the University of Iowa Institutional Review Board.

## Neuropsychological assessment

A set of standardized neuropsychological tests was used to assess neuropsychological outcomes. To account for age-related effects, all test scores were converted to age-adjusted z-scores using the mean and standard deviation from published population normative data. We determined the behavioral domain that each test assessed, as described in *Neuropsychological assessment* (*Lezak et al., 2012*). We compared test outcomes between thalamus and comparison patients using the non-parametric randomized permutation test. Each test creates an empirical null distribution of no group difference between patients by randomly permuting group membership (thalamus or comparison patients) of each test score while keeping the number of patients in each group constant. We reported significant results after correcting for multiple comparisons using the Bonferroni correction (10 neuropsychological tests, $p < 0.005$).

We used TMT Part B test scores to assess executive function. Stimuli in TMT Part B consist of both numbers and letters scattered on a page, and patients are asked to use a pencil to connect circles between them, in an alternating sequence (i.e., 1-A-2-B-3-C, etc.) as quickly as possible. TMT Part B is considered to be a test of control-related functions that include working memory and cognitive flexibility (*Crowe, 1998*; *Kortte et al., 2002*). In contrast, TMT Part A is an easier test, which requires patients to use a pencil to connect 25 circled numbers in numeric order as quickly as possible. Part A is thought to test psychomotor functions (*Bowie and Harvey, 2006*).

The BNT and COWA tests were used to assess verbal naming and verbal fluency functions. The BNT consists of 60 line drawings depicting objects for subjects to name (*Tombaugh and Hubley, 1997*). COWA presents a set of letters; subjects are then are asked to say as many words as they can think of that begin with a given letter (*Ruff et al., 1996*).

Various components from RAVLT were used to assess learning and memory functions (*Vakil and Blachstein, 1993*). Subjects were presented with a 15-word list and then asked to repeat as many words as they could over five recall trials. They were then given a delayed recognition and a delayed recall test after a 30 min delay. Delayed recall and delayed recognition test scores were calculated using correct recalls/recognitions minus the number of false recalls/recognitions. We did not administer the 'Trial B' interference procedure. We assessed the immediate learning capacity from the score of the first trial (RAVLT first trial, also known as immediate recall), the cumulative of learning outcome (summing the scores across the five trials; RAVLT learning), and long-term verbal memory (RAVLT delayed recall and RAVLT delayed recognition).

We used the Rey-Osterrieth Complex Figure Test to assess visuospatial memory and construction (*Fastenau et al., 1999*). The Rey-Osterrieth figure was presented, and subjects were asked to copy the figure onto a blank paper (the copy trial). After a 30 min delay, subjects were asked to recall and draw the figure from memory (the delayed recall trial). We did not administer the immediate recall test. A standardized scoring system was used to assess the accuracy of a subject's copy and recall performances (*Meyers and Meyers, 1995*).

## Anatomical analysis of lesion location

The anatomic location and spatial extent of each lesion was determined using available T1, T2, and computed tomography data. Because all patients were selected from the Iowa Neurological Patient Registry, which has been continuously enrolling patients for decades, imaging data were acquired

using a variety of sequences. For T1 and T2 data, most images were acquired with 0.9375 × 0.9375 × 1.5 mm³ or 1 × 1 × 1 mm³ resolution; for computed tomography data, data were acquired with 0.94 × 0.94 mm² in-plane resolution, with slice thickness ranging from 2 to 5 mm. All lesions were manually traced by trained technicians and reviewed by a board-certified neurologist (co-author ADB), who was blinded to neuropsychological outcome results. Lesion masks were transformed to the MNI International Consortium for Brain Mapping (ICBM) 152 Nonlinear Asymmetrical template, version 2009 c, using a procedure that we reported previously (*Hwang et al., 2020*). Briefly, we used a high-deformation, non-linear, enantiomorphic, registration procedure from the Advanced Normalization Tools (*Avants et al., 2009*; *Brett et al., 2001*; *Nachev et al., 2008*). This high-deformation, non-linear registration procedure allows local deformation to account for differences in size and shape between brain structures. We used enantiomorphic normalization to insert voxel intensities from the non-damaged homologue of the lesion site in place of the manually defined lesion mask to improve transformation accuracy. After transformation, lesion masks went through a second round of manual editing as needed, to ensure that the anatomical borders of the lesion were accurately represented on the template atlas. The Morel atlas was used to determine the location of different thalamic nuclei (*Krauth et al., 2010*). This atlas identifies human thalamic nuclei based on cyto- and myelo-architecture information in stained slices from five postmortem human brains, and further transformed to the template space.

## Functional connectivity and network hub analyses

The normative functional connectome dataset consisted of resting-state fMRI data from 235 subjects (mean age = 21.7 years, SD = 2.87 years, age range = 19–27, 131 males). Data from these subjects were acquired as part of the Brain Genomics Superstruct Project (*Holmes et al., 2015*), which we have previously used to map the hub properties of the thalamus (*Hwang et al., 2017*). For each norma-tive subject, two 6-min runs of fMRI data were collected using a gradient-echo echo-planar imaging sequence (repetition time [TR] = 3000 ms, echo time [TE] = 30 ms, flip angle = 85 degrees, 3 mm³ isotropic voxels with 47 axial slices). We replicated our results using 62 subjects from an independent dataset from the Nathan Kline Institute-Rockland Sample (*Nooner et al., 2012*). For the replication dataset, 9 min and 35 s of resting-state fMRI data were collected for each subject using with the following parameters: TR = 1400 ms, TE = 30 ms, multiband factor = 4, flip angle = 65 degrees, 2 mm³ isotropic voxels with 64 axial slices.

To prepare fMRI data for connectivity and network analyses, brain images were segmented into different tissue classes (white matter [WM], GM, and cerebrospinal fluid [CSF]) using FSL's FAST software which helps co-register functional and anatomical data using boundary-based registration algorithm. We used rigid body motion correction to correct for head motion. T1 data were then spatially normalized to the MNI-152 space using the same high-deformation, non-linear function from the Advanced Normalization Tools that we used to transform lesion masks into the MNI space. We performed anatomical CompCor nuisance regression to further reduce non-neural noise (*Behzadi et al., 2007*). The close physical proximity between the thalamus and high-noise regions, such as the ventricles, could result in blurring the fMRI signal. To minimize this confound we further regressed out the mean signals from CSF, WM, and GM that were within five voxels (10 mm) from the thalamus. Importantly, no spatial smoothing was performed. After regression, data were bandpass-filtered (0.009–0.08 Hz).

To calculate thalamocortical functional connectivity, we used the Morel atlas (*Krauth et al., 2010*) to define the thalamus (2227 2 mm³ voxels included in the atlas, registered to the MNI template), and calculated Pearson correlations between thalamic voxels and 400 cortical ROIs (*Schaefer et al., 2018*). Note that no correlations were calculated between thalamic voxels. This procedure resulted in a 2227 (thalamus voxel) × 400 cortical ROI matrix. To estimate the connector hub property of each thalamus voxel, we calculated its PC value across a range of density thresholds of this thalamocortical matrix (density = 0.01–0.15) and averaged across thresholds. The PC value of thalamus voxel i is defined as:

$$PC = 1 - \sum_{s=1}^{N_M} \left( \frac{K_{is}}{K_i} \right)^2,$$

where $K_i$ is the sum of total functional connectivity weight for voxel i, $K_{is}$ is the sum of functional connectivity weight between voxel i and the cortical network s, and NM is the total number of

networks. To perform this calculation, we assigned the 400 cortical ROIs to seven cortical functional networks including FP, DF, CO, DA, limbic, SM, and visual networks (*Schaefer et al., 2018*; *Yeo et al., 2011*). If a thalamic voxel acts as a connector hub for cortical functional networks, it should exhibit functional connectivity uniformly distributed with cortical networks, and its PC value will be close to 1; otherwise, if its functional connectivity is concentrated within a specific cortical network, its PC value will be close to 0 (*Gratton et al., 2012*; *Guimerà and Nunes Amaral, 2005*).

In addition to estimating PC values for thalamic voxels, we also estimated PC values for whole-brain GM voxels using the same approach (*Reber et al., 2021*). Briefly, we defined GM voxels using the MNI atlas (*Fonov et al., 2009*), and calculated Pearson correlations between signals extracted from every GM voxel (4 mm$^3$ resolution, total 18,166 voxels). To avoid bias, we did not calculate correlations between voxels within the same ROI (*Schaefer et al., 2018*). We assigned each GM voxel to one of the seven functional networks in the cortex (*Yeo et al., 2011*) or its corresponding subcortical parcellation (*Buckner et al., 2011*; *Choi et al., 2012*). PC was calculated for a range of thresholds (density = 0.01–0.15) and averaged across thresholds.

## Neurosynth meta-analyses

We used the Neurosynth database (*Yarkoni et al., 2011*) to identify brain regions associated with putative cognitive processes that are assessed by the TMT Part B, BNT, COWA, RAVLT delayed recall, and RAVLT delayed recognition tests. Neurosynth performs automated meta-analyses on a large fMRI corpus, with more than 14,000 fMRI studies included in the database. We quarried the following terms, 'executive' (for TMT Part B), 'naming' (for BNT), 'fluency' (for COWA), 'recall' (for RAVLT delayed recall), and 'recognition' (for RAVLT delayed recognition). Neurosynth then performed an association test for each term, to identify voxels more consistently reported in fMRI studies that contained the queried term than studies that did n't contain the term. The resulting maps depicts putative brain systems whose activities are associated with the searched term.

## Estimating the density of matrix cells

Procedures used to estimate the densities of matrix and core projection cells were previously published (*Müller et al., 2020*). Briefly, we obtained spatial maps (in MNI-152 space) of mRNA expression levels for PVALB and CALB1 proteins provided by the Allen Human Brain Atlas (*Gryglewski et al., 2018*). These proteins have been previously shown to delineate matrix and core thalamic projection cells (*Jones and Hendry, 1989*). Voxel-wise mRNA levels were first normalized and transformed into z-scores across all voxels within the thalamus, and the voxel-wise distributions of these normalized values were compared between lesion sites. A difference score was also calculated to identify thalamic voxels with higher densities of matrix cells.

## Code and data availability statement

Functional connectivity analyses utilized publicly available datasets (*Holmes et al., 2015*; *Nooner et al., 2012*). Code and de-identified neuropsychological assessment outcome data can be accessed at https://github.com/kaihwang/LTH; *Hwang, 2021* copy archived at swh:1:rev:fdcca3fd575911fa42fc79bccc468c2f9f6e6153.

## Acknowledgements

KH was supported by National Institutes of Health R01MH122613. JMS was supported by National Health and Medical Research Council GNT1156536 and National Institutes of Health RO1MH117772. DT was supported by National Institutes of Health P50 MH094258 and the Kiwanis Neuroscience Research Foundation. ADB was supported by National Institutes of Health R01NS114405 and R21MH120441. Portions of this work were conducted on an MRI instrument funded by National Institutes of Health S10OD025025-01. The content is solely the responsibility of the authors and does not represent the official views of the National Institutes of Health.

## Additional information

### Funding

| Funder | Grant reference number | Author |
|---|---|---|
| National Institutes of Health | R01MH122613 | Kai Hwang<br>Daniel Tranel<br>Aaron Boes |
| National Institutes of Health | RO1MH117772 | James M Shine |
| National Institutes of Health | P50MH094258 | Daniel Tranel |
| Kiwanis Neuroscience Research Foundation | | Daniel Tranel |
| National Institutes of Health | R01NS114405 | Aaron Boes |
| National Institutes of Health | R21MH120441 | Aaron Boes |
| National Health and Medical Research Council | GNT1156536 | James M Shine |

The funders had no role in study design, data collection and interpretation, or the decision to submit the work for publication.

### Author contributions

Kai Hwang, Conceptualization, Data curation, Formal analysis, Funding acquisition, Investigation, Methodology, Resources, Software, Supervision, Validation, Visualization, Writing - original draft, Writing – review and editing; James M Shine, Formal analysis, Methodology, Resources, Writing - original draft, Writing – review and editing; Joel Bruss, Data curation, Formal analysis, Methodology, Software, Writing – review and editing; Daniel Tranel, Data curation, Funding acquisition, Project administration, Resources, Supervision, Writing – review and editing; Aaron Boes, Data curation, Formal analysis, Funding acquisition, Investigation, Methodology, Project administration, Resources, Supervision, Writing - original draft, Writing – review and editing

### Author ORCIDs

Kai Hwang http://orcid.org/0000-0002-1064-7815
James M Shine http://orcid.org/0000-0003-1762-5499

### Ethics

All participants gave written informed consent, and the study was approved by the University of Iowa Institutional Review Board (protocol #200105018).

### Decision letter and Author response

Decision letter https://doi.org/10.7554/eLife.69480.sa1
Author response https://doi.org/10.7554/eLife.69480.sa2

## Additional files

### Supplementary files

• Transparent reporting form
• Supplementary file 1. Demographic data for all patients.

### Data availability

We have made all code and lesion-derived measures used in the manuscript freely available on github (https://github.com/kaihwang/LTH, (copy archived at swh:1:rev:fdcca3fd575911fa42fc79b-ccc468c2f9f6e6153)), including neuropsych assessment outcome, derivatives from lesion analyses, data used for functional connectivity analyses, and mRNA expression analyses. Functional connectivity

analyses utilized publicly available datasets (Holmes et al., 2015; Nooner et al., 2012). The only data that we cannot post without restrictions are each patient's clinical MRI data and lesion data. Patients were enrolled into the Iowa Lesion Patient Registry the past few decades, and most did not consent to post their clinical MRI data publicly. To gain access to those data, the interested party will have to contact the PI of the lesion registry, Dr. Dan Tranel, and the corresponding author of this project, Dr. Kai Hwang. The user will require to sign a data use agreement. This institutional policy was designed to ensure the appropriate use of the data for academic and not commercial purposes. A study plan of the proposed research will have to be submitted, and we will work with the interested party to obtain the necessary IRB approval from both institutions.

The following previously published datasets were used:

| Author(s) | Year | Dataset title | Dataset URL | Database and Identifier |
| --- | --- | --- | --- | --- |
| Homes et al | 2015 | Brain Genomics Superstruct Project | https://www.nature.com/articles/sdata201531 | Brain Genomics Superstruct Project initial data release with structural, functional |
| Nooner et al | 2012 | NKI-Rockland sample | http://fcon_1000.projects.nitrc.org/indi/enhanced/ | The enhanced Nathan Kline Institute-Rockland Sample, NKI-RS |

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
