## [Decision Letter]

**Acceptance summary:**

Hwang et al., address the important question about whether specific thalamic sub-regions serve as essential "hubs" for interconnecting diverse cognitive processes. Using a group of patients with isolated thalamic lesions (n=20), and a group of size-matched lesions outside the thalamus (n=42), they report that lesions to the anterior-medio-dorsal thalamus are most likely to cause widespread cognitive deficit. Evidence from existing task-based and resting-state fMRI data sets, as well as data sets on gene expression further provide evidence the importance of these regions as a network hub for domain general cognitive functions.

**Decision letter after peer review:**

Thank you for submitting your article "Neuropsychological evidence of multi-domain network hubs in the human thalamus" for consideration by *eLife*. Your article has been reviewed by 2 peer reviewers, and the evaluation has been overseen by a Reviewing Editor and Timothy Behrens as the Senior Editor. The following individual involved in review of your submission has agreed to reveal their identity: Nico Dosenbach (Reviewer #2).

Essential revisions:

A number of issues that may influence the validity and interpretation of the lesion analysis were raised in the review and the subsequent discussion and should be addressed through additional analyses, or at least clear discussed in the manuscript.

1) One concern that the primary "multi-domain" overlap site is also the primary site of overlap in general across thalamic lesion cases (Figure 2A). Could this uneven sampling of lesion locations bias the results, especially given the low N? For examples, if the posterior thalamus (pulvinar) was also a "hub" region would the authors have had the power to detect such a deficit?

2) Many lesions appear to also involve white matter components. Given that white matter damage may have systematically different consequences as gray matter damage, it may be important to control for these characteristics.

3) The general use of cortical lesions as control raises the question of whether some of these cortical locations may also be "hub" regions. It would be useful to determine whether or not hub locations in the cortex and thalamus show similar properties.

4) It appears from Figure 2B that the patients with multi-domain impairment also simply are more impaired overall. So can the authors really distinguish between the mild / severe deficits on the one hand and specific / global deficits on the other? If the authors wish to claim that it is specifically the profile of the deficit, rather than the strength of the deficit, additional analysis are required. If not, it should be clarified that the two cannot be distinguished based on the current data.

5) Is it possible that the lesions that lead to "hub" deficits simply involve more thalamic subregions than lesions that do not? Could the inter-subject variability of the organization of these networks possible influence the result?

Specific comments of the individual reviewers are attached below and also should be addressed.

*Reviewer #1 (Recommendations for the authors):*

1. As noted in the first comment above, I would suggest that the authors address the small size of their sample either through a more directed approach (testing hub vs. non-hub thalamic locations), replication (perhaps another lesion database, such as from the Corbetta lab would offer this opportunity), and/or more extended discussion about the limitations of testing this question in a small sample.

2. In response to comment 2 above, I would recommend that the authors conduct additional analyses to compare the amount of white matter damage seen with thalamic hub/non-hub lesions and comparison participants.

3. In response to question 3 above, I would recommend separating hub and non-hub locations of the cortex for comparison to the thalamic locations, perhaps conducting a similar overlap analysis on the cortical locations to confirm that the current approach works in a better powered group.

4. In response to comment 4, I would recommend the authors conduct the additional control analyses listed in that location.

5. The results text says that thalamus lesion patients did worse than comparisons on TMT Part B, but reports a higher Z statistic – is this an error? The values look lower in the figure.

6. I did not understand the purpose of the correlations shown in Figure 3, especially given the small sample of patients with thalamic lesions. I would recommend removing or clarifying this analysis further. Would differences in the multi-domain impairment score (y-axis) be sufficient to make the primary point?

7. Why are only 4 of the thalamic hub lesion patients shown in Supp. Figure 1? Given the small sample size, I think it would be appropriate to show all of the thalamic hub and non-hub lesions.

*Reviewer #2 (Recommendations for the authors):*

A few suggestions that might further improve the manuscript.

It might be helpful for readers to list the full names and the between group statistical testing results in table format instead of describing them all in text.

For Figure 1 it might be helpful to enlarge panel A relative to panel B for easier viewing of the very important lesion maps. For visually parsing the horizontal slices through the thalamus, it might also be helpful to give larger white spaces, similar to the layout in Figure 2.

In Figure 2A the color scale for the overlap panels on the right seems a little curious because it includes one cool color and the rest are all hot colors, might be more standard to use all hot colors for that one.

Please give the center of mass ROI coordinates for the anterior-medio-dorsal thalamus hubs in MNI or Tailarach.

In Figure 3 the gray comparison patient dots are a little too faint, consider darkening them, so they're more easily visible.

Line 345 typo: postmoterm

Line 357 typo: analogues

For Figure 7 it might be helpful to also include an overlap quantification, maybe with a Dice coefficient.

In the Discussion section it might be helpful to discuss the relative 'hubness' of thalamic hubs vs. cortical hubs. Which are more hub-like? Do you think they are fundamentally different in some way? A lot of neuroimaging research only focuses on the cortex, so are the cortical hubs examples of looking under the light post for the keys or are they just as strong as the thalamic hubs?

I really like this very tempered statement in the Discussion:

"what is the function that is implemented by a thalamic hub that allows it to be broadly involved in cognition? Our findings constrain the possible answers-it must be processes that are not specific to a single cognitive function but can be generalized across domains"

Any chance you speculate a little more what types of thalamic processes could be general to all the affected domains. What might it be? What computations does the anterior-medio-dorsal thalamus perform?

The CALB1/PVALB result is super interesting, I wanted to see it discussed even a little more.

---

## [Author Response]

Essential revisions:A number of issues that may influence the validity and interpretation of the lesion analysis were raised in the review and the subsequent discussion and should be addressed through additional analyses, or at least clear discussed in the manuscript.1) One concern that the primary "multi-domain" overlap site is also the primary site of overlap in general across thalamic lesion cases (Figure 2A). Could this uneven sampling of lesion locations bias the results, especially given the low N? For examples, if the posterior thalamus (pulvinar) was also a "hub" region would the authors have had the power to detect such a deficit?

We acknowledge that the uneven sampling of thalamic lesion locations is a limitation of our study. As reviewer one pointed out, focal lesions in the thalamus are relatively rare, and we had to exclude patients with thalamic lesions that expanded to nearby white matter or cortical structures. It is relatively difficult to obtain a high-powered sample with even sampling of lesion locations throughout the thalamus.

If we had enough patients with posterior thalamic lesions, would we be able to identify it as a multi-domain hub? Studies that focused on mapping “hub” like integrative subregions in the human thalamus have found the dorsal thalamus, the anterior-medial, and the posterior-medial thalamus to exhibit the strongest hub properties (Cole et al., 2010, Hwang et al., 2017, Greene et al., 2020). These findings are consistent with our results demonstrating that lesions to the anterior-medio-dorsal thalamus are associated with behavioral impairments across multiple cognitive domains. Given that functional connectivity studies also found strong hub property in the posterior-medial thalamus, one prediction would be that patients with focal lesions in the posterior-medial thalamus should also exhibit behavioral impairments across multiple cognitive domains.

A competing predication is that the posterior-medial thalamus is not a multi-domain hub. We found that the CALB1 protein, a marker of “matrix” projection cells (Jones et al., 2001), is more densely expressed in the anterior but not the posterior thalamus (the current study and Müller et al., 2020). Matrix cells project to segregated cortical regions, a connectivity motif that is characteristic of brain network hubs. A higher concentration of CALB1 in the anterior but not the posterior thalamus would predict that the anterior-medial, but not the posterior-medial thalamus, is a multi-domain hub. Not finding multi-domain hubs in the posterior-medial thalamus would further suggest that participation coefficient, a graph metric that we and many others used for mapping brain network hubs, may need to be paired with other data and measures to identify multi-domain hubs. The current study cannot adjudicate between these two predictions, and we have updated the Discussion section (line 561) to acknowledge this limitation.

2) Many lesions appear to also involve white matter components. Given that white matter damage may have systematically different consequences as gray matter damage, it may be important to control for these characteristics.

We would like to clarify that we excluded thalamic patients with substantial white matter damage, for example lesions to the adjacent internal capsule. As reviewer 1 pointed out, many comparison patients had white matter damage, and we agree that we should control for effects related to white matter lesions. Out of the original 42 comparison patients (to control for lesion size) and 320 expanded comparison patients (to control for average impairment severity), only 4 patients did not have white matter lesions, therefore we were not able to completely exclude patients with white matter lesions. Instead, we restricted our analyses by only including comparison patients with predominantly gray matter lesions (>80% of total lesion volume). This selection reduced our sample size to 21 comparison patients (Figure 1) and 145 expanded comparison patients (Figure 3 and Figure 6). We have updated all analyses with this updated list of comparison patients, and the results were consistent with our original analysis. Specifically, we found that patients with thalamic lesions performed significantly worse than the new list of comparison patients on the following tests: TMT Part B, BNT, RAVLT delayed recall, and RAVLT delayed recognition (Table 1). These tests covered the executive function, verbal fluency, and memory domains. Thalamus patients also exhibited more global deficits, exhibiting impairments on higher numbers of cognitive domains. These updated results suggest that our findings likely cannot be explained by the larger volumes of white matter damage in comparison patients.

3) The general use of cortical lesions as control raises the question of whether some of these cortical locations may also be "hub" regions. It would be useful to determine whether or not hub locations in the cortex and thalamus show similar properties.

Lesions to cortical hubs are known to be associated with behavioral impairments across multiple cognitive domains (Warren et al., 2014, Reber et al., 2021). To replicate prior findings and validate our approach, we applied the method we used for identifying multi-domain lesion sites in the thalamus to map lesion sites with similar behavioral profiles in comparison patients. Given the limited lesion coverage of the 21 comparison patients matched in overall lesion volumes, we expanded the comparison group and identified 145 expanded comparison patients with lesions that predominantly involved the gray matter (>80% of total lesion column). Using the same approach that we used to locate multi-domain hubs in the thalamus, we then examined the profile of neuropsychological outcomes of each comparison patient, and classified patients into two sub-groups: those with (n = 58) and without (n=87) multi-domain impairment. We then compared the hub properties of lesion sites between these two groups of comparison patients. To assess the hub property of each comparison lesion, we calculated the participation coefficient (PC) value of every gray matter voxel using a whole-brain, voxel-wise mapping approach that we previously published (Reber et al., 2021; Line 771 in the methods section). To reduce computation burden, it was necessary to reduce the spatial resolution to 4 mm^3^ voxel for the whole-brain voxel-wise analysis. We then compared the lesion sites’ PC values between comparison patients with and without multi-domain impairment.

This additional analysis validated our approach (Figure 6). First, we found that comparison lesions associated with multi-domain impairment were primarily located in lateral frontal and posterior parietal associative regions known to exhibit strong hub properties (Figure 6A). In contrary, cortical lesions associated with impairment in only one cognitive domain primarily overlapped with primary cortices. We then compared the distribution of voxel-wise PC values between the multi-domain and single-domain lesion sites, and found that voxels in the multi-domain lesion sites had on average higher PC values when compared to those in the single domain lesion sites (Figure 6C; Kolmogorov-Smirnov d = 0.11, p < 0.001; replication dataset: Kolmogorov-Smirnov d = 0.099, p < 0.001).

We then compared the hub properties between thalamic lesions and comparison lesions associated with multi-domain impairment (Figure 6D), and found that comparison lesions had a wider distribution of PC values, likely because comparison lesions included in Figure 6 were significantly larger than thalamic lesions (comparison lesions: mean=36440 mm^3^, SD=33078 mm^3^, thalamic lesions: mean = 1559 mm^3^, SD = 1312 mm^3^, randomize permutation test p < 0.001). Given the difference in lesion size and distribution, we did not statistically compare PC values between lesions from thalamic and expanded comparison patients. We found no statistically significant difference in the multiple-domain impairment score between thalamic and expanded comparison patients with hub lesions (Figure 6E, randomized permutation test p = 0.17), suggesting that these two groups of patients were both impaired on multiple cognitive domains.

Taken together, these findings support the approach we used for linking lesions to thalamic hubs with multi-domain impairment, as this approach is also effective in deriving the same association for lesions to hub regions outside the thalamus in a much larger cohort. We have added this analysis to the revised manuscript.

4) It appears from Figure 2B that the patients with multi-domain impairment also simply are more impaired overall. So can the authors really distinguish between the mild / severe deficits on the one hand and specific / global deficits on the other? If the authors wish to claim that it is specifically the profile of the deficit, rather than the strength of the deficit, additional analysis are required. If not, it should be clarified that the two cannot be distinguished based on the current data.

We agree that it is important to examine the relationship between global impairment versus the average impairment severity. This issue was also raised during the initial editorial assessment, which we responded by including Figure 3 to illustrate the relationship between these two variables. Specifically, on the X axis, we plotted the “average impairment score” by averaging the normalized z-scores across all 10 neuropsychological tests, where a negative z-score represents more severe deficit. On the Y axis, we plotted the “multi-domain impairment score” by summing the number of tests with significant behavioral deficits (defined as Z < 1.645), where a positive score indicates more global impairment. Given that larger lesions were associated with more severe behavioral impairment, we further adjusted these scores by controlling for the effects of lesion size via linear regression.

Several observations can be made from Figure 3. First, there is a significant linear relationship between average severity and global deficits (b = -1.14, R^2^=0.15, p <0.001), indicating that we cannot completely dissociate the impact of average behavioral impairment severity from global deficits. Second, despite this correlation, several thalamus patients with anterior-medio-dorsal lesions exhibited similar levels of average impairment (X axis -0.3 to 0.4) when compared to thalamic patients without anterior-medio-dorsal lesions. Third, we fitted a linear regression line to the comparison patients that had average impairment scores similar to thalamus patients (X axis maximum = 1.39, minimum = -1.59; grey regression line in Figure 3), and found that most thalamus patients with anterior-medio-dorsal lesions exhibited higher multi-domain impairment score when compared to comparison patients with similar levels of average impairment. Finally, a closer examination of results presented in Figure 2B suggests that there is specificity in the type of behavioral impairment observed. For example, across all patients, the Complex Figure tests were less impaired, potentially because of intact functional specialization in our thalamus patient group.

Therefore, it is clear from our results that lesions associated with similar levels of overall severity can have varying levels of global deficit. Lesion locations and the hub properties of the lesion locations can partially explain the specificity and globality of behavioral deficits. Lesions to the left anterior-medio-dorsal thalamus were not merely associated with more severe behavioral impairment, but also associated with impairment across multiple cognitive domains to a greater extent than would be expected from lesions in other brain regions.

Nevertheless, our results indicate that we cannot completely dissociate average severity from the globality of behavioral deficits. In the Discussion section, we highlighted several potential explanations (line 520). Specifically, it is possible that a set of latent, common cognitive processes are necessary for different neuropsychological tests. Anterior-medio-dorsal thalamic lesions may affect these latent processes, and impact behavior across domains without specificity. For example, the supraordinate, task-general “multiple-demand” or task control (frontoparietal and cingulo-opercular) networks may be necessary for optimal performance across different tests, and these networks were affected in patients with anterior-medio-dorsal thalamus (“diaschisis”), manifesting in a more severe and globally impaired behavioral profile. We have updated our Discussion section to discuss these nuances (line 520).

5) Is it possible that the lesions that lead to "hub" deficits simply involve more thalamic subregions than lesions that do not? Could the inter-subject variability of the organization of these networks possible influence the result?

Using the Morel thalamus nuclei atlas (Krauth et al., 2010), we compared the number of nuclei involved in lesions associated with and without multi-domain impairment, and found no significant difference (multiple domain lesions: mean = 4.9 nuclei, SD = 1.89; single domain lesions: mean = 5.92 nuclei, SD = 2.7; randomized permutation p = 0.39). Using a thalamus functional parcellation atlas we previously published (Hwang et al., 2017), we also did not find significant differences in the number of network parcellations between these lesions sites (multiple domain lesions: mean = 3.25 parcels, SD = 1.87; single domain lesions: mean = 3.88 parcels, SD = 2.59; randomized permutation p = 0.62). We replicated this finding using an independent thalamic parcellation from Thomas Yeo (https://twitter.com/bttyeo/status/1248992127985434624; multiple domain lesions: mean = 4.55 parcels, SD = 1.44; single domain lesions: mean = 3.95 parcels, SD = 1.93; randomized permutation p = 0.21). These results indicate that thalamus lesions associated with more global deficits did not involve more thalamic subregions than lesions that did not.

Specific comments of the individual reviewers are attached below and also should be addressed.Reviewer #1 (Recommendations for the authors):1. As noted in the first comment above, I would suggest that the authors address the small size of their sample either through a more directed approach (testing hub vs. non-hub thalamic locations), replication (perhaps another lesion database, such as from the Corbetta lab would offer this opportunity), and/or more extended discussion about the limitations of testing this question in a small sample.

We examined the lesion location from 130 patients from an independent dataset, and unfortunately we did not find enough cases of focal thalamic lesions in that dataset. Most lesion registries do not consistently recruit subcortical patients, and this is a clear limitation. We have more explicitly discussed the limitations from the relatively small sample size (line 561).

2. In response to comment 2 above, I would recommend that the authors conduct additional analyses to compare the amount of white matter damage seen with thalamic hub/non-hub lesions and comparison participants.

Please see our response to essential revisions point #2.

3. In response to question 3 above, I would recommend separating hub and non-hub locations of the cortex for comparison to the thalamic locations, perhaps conducting a similar overlap analysis on the cortical locations to confirm that the current approach works in a better powered group.

Please see our response to essential revisions point #3.

4. In response to comment 4, I would recommend the authors conduct the additional control analyses listed in that location.

Please see our response to essential revisions point #5.

5. The results text says that thalamus lesion patients did worse than comparisons on TMT Part B, but reports a higher Z statistic – is this an error? The values look lower in the figure.

We apologize for the confusion. For the trail making task, longer reaction time indicates more severe impairment. To facilitate the comparison between TMT and other tasks, we “inverted” the TMT z-scores (both part A and part B) for all figures so negative Z statistics would indicate worse performance on all tasks. We have made that clear in the figure legend, and updated the main manuscript so the reporting is consistent with the figure.

6. I did not understand the purpose of the correlations shown in Figure 3, especially given the small sample of patients with thalamic lesions. I would recommend removing or clarifying this analysis further. Would differences in the multi-domain impairment score (y-axis) be sufficient to make the primary point?

Please see our detailed response to essential revisions point #4 above. We included figure 3 to illustrate the relationship between average behavioral impairment severity versus global deficits across cognitive domains. Importantly, we found that when matched with overall behavioral impairment severity, thalamus patients with lesions to the anterior-medio-dorsal thalamus exhibited more global multi-domain deficits when compared to comparison patients.

7. Why are only 4 of the thalamic hub lesion patients shown in Supp. Figure 1? Given the small sample size, I think it would be appropriate to show all of the thalamic hub and non-hub lesions.

We have updated the supplementary figure to include all thalamus patients.

Reviewer #2 (Recommendations for the authors):A few suggestions that might further improve the manuscript.It might be helpful for readers to list the full names and the between group statistical testing results in table format instead of describing them all in text.

We have included a new Table 1 to list the full names of the neuropsychological assessments, the descriptive statistics, and the randomized permutation p values.

For Figure 1 it might be helpful to enlarge panel A relative to panel B for easier viewing of the very important lesion maps. For visually parsing the horizontal slices through the thalamus, it might also be helpful to give larger white spaces, similar to the layout in Figure 2.

We have updated Figure 1 and enlarged the lesion location panel as suggested. We have also enlarged the white spaces between panels.

In Figure 2A the color scale for the overlap panels on the right seems a little curious because it includes one cool color and the rest are all hot colors, might be more standard to use all hot colors for that one.

We have updated the color scheme in the right panel of Figure 2A to use a monochrome color scale.

Please give the center of mass ROI coordinates for the anterior-medio-dorsal thalamus hubs in MNI or Tailarach.

We now report the center coordinate anterior-medio-dorsal thalamus lesion site on line 202 (MNI: -7, 10, 8).

In Figure 3 the gray comparison patient dots are a little too faint, consider darkening them, so they're more easily visible.

We have reduced the transparency of the gray dots in Figure 3 to make them more visible.

Line 345 typo: postmotermLine 357 typo: analogues

Thank you for pointing them out, these have been fixed.

For Figure 7 it might be helpful to also include an overlap quantification, maybe with a Dice coefficient.

We calculated the Dice coefficient to compare the spatial overlap between the CALB1 expression maps and lesion masks. We found the CALB1 expression map to be more similar to the multi-domain lesion sites (Dice coefficient = 0.49), and less similar to the single-domain lesion site (Dice coefficient = 0.007). These results are now reported on line 421.

In the Discussion section it might be helpful to discuss the relative 'hubness' of thalamic hubs vs. cortical hubs. Which are more hub-like? Do you think they are fundamentally different in some way? A lot of neuroimaging research only focuses on the cortex, so are the cortical hubs examples of looking under the light post for the keys or are they just as strong as the thalamic hubs?

Consistent with prior studies, we found that thalamic hubs exhibited integrative connectivity pattern with multiple cortical functional networks (Figure 5B and Figure 7A), and had high PC values comparable to cortical hubs (Figure 6D, Hwang et al., 2017). Critically, small lesions to thalamic hubs were associated with broad, global behavioral deficits similar to lesions to cortical hubs (Figure 6E). It is important to note that cortical lesions tend to be much larger and almost invariably involve at least some underlying white matter. While we do not have enough data to conclusively determine which structure is more “hub” like, our results do suggest that a 500 mm^3^ lesion in the thalamus will likely have much greater behavioral impact compared to a 500 mm^3^ cortical lesion. It remains to be tested whether a small focal lesion to critical cortical hubs can have similar behavioral effects that are comparable to a small but significant thalamic hub lesion. Our results, coupled with other previous studies, definitely suggest that whole-brain neuroimaging studies should not exclude subcortical structures.

I really like this very tempered statement in the Discussion:"what is the function that is implemented by a thalamic hub that allows it to be broadly involved in cognition? Our findings constrain the possible answers-it must be processes that are not specific to a single cognitive function but can be generalized across domains"Any chance you speculate a little more what types of thalamic processes could be general to all the affected domains. What might it be? What computations does the anterior-medio-dorsal thalamus perform?

We appreciate the reviewer for raising this question. In the Discussion section (line 520) we discussed several potential mechanisms. One possibility is the thalamic hubs maintain an optimal modular organization of functional brain networks and promote both segregated and integrative functions that are necessary for multiple cognitive functions. Another possibility is that the anterior-medio-dorsal thalamus is part of a core “multiple-demand” network, or interact closely with several control-related networks (cingulo-opercular and frontoparietal), and therefore is involved in task-general, control processes that are necessary for tasks across different cognitive domains. The third possibility is that the thalamus is involved in modulating cortical evoked responses and promoting inter-regional communication. This possibility is supported by findings from animal models, demonstrating that in rodents inactivating the thalamus diminished cortical evoked responses and decreased cortico-cortical communication. We hypothesize that these mechanisms are not domain-specific, and different cognitive functions may recruit one or more of these general mechanisms depending. Lesions to thalamic hubs affects the integrity of these mechanisms, which in turn broadly affects multiple cognitive functions.

The CALB1/PVALB result is super interesting, I wanted to see it discussed even a little more.

We analyzed the density of CALB1/PVALB expression to identify the relative concentration of “matrix” projection cells in different thalamic subregions. Calbindin-rich matrix cells project diffusively to the superficial layers of the cerebral cortex, unconstrained by functional borders between cortical regions. The distributed projection pattern of matrix cells suggests that it may be the anatomical connectivity substrate that allows the anterior-medio-dorsal thalamus to carry out its connector hub functions. In the discussion, we clarified the connection between matrix cells and the connectivity organization of thalamic hubs (line 489). We then discussed several hypothetical network functions of thalamic hubs, including maintaining the modular organization of functional systems, interacting with task control networks, modulating evoked responses and inter-regional communication (line 520).

References

Jones, E. G. (2001). The thalamic matrix and thalamocortical synchrony. *Trends in neurosciences*, *24*(10), 595-601.

Müller, E. J., Munn, B., Hearne, L. J., Smith, J. B., Fulcher, B., Arnatkevičiūtė, A., … and Shine, J. M. (2020). Core and matrix thalamic sub-populations relate to spatio-temporal cortical connectivity gradients. *NeuroImage*, *222*, 117224.

Cole, M. W., Pathak, S., and Schneider, W. (2010). Identifying the brain's most globally connected regions. *Neuroimage*, *49*(4), 3132-3148.

Warren, D. E., Power, J. D., Bruss, J., Denburg, N. L., Waldron, E. J., Sun, H., … and Tranel, D. (2014). Network measures predict neuropsychological outcome after brain injury. *Proceedings of the National Academy of Sciences*, *111*(39), 14247-14252.

Reber, J., Hwang, K., Bowren, M., Bruss, J., Mukherjee, P., Tranel, D., and Boes, A. D. (2021). Cognitive impairment after focal brain lesions is better predicted by damage to structural than functional network hubs. *Proceedings of the National Academy of Sciences*, *118*(19).

Greene, D. J., Marek, S., Gordon, E. M., Siegel, J. S., Gratton, C., Laumann, T. O., … and Dosenbach, N. U. (2020). Integrative and network-specific connectivity of the basal ganglia and thalamus defined in individuals. *Neuron*, *105*(4), 742-758.

Hwang, K., Bertolero, M. A., Liu, W. B., and D'Esposito, M. (2017). The human thalamus is an integrative hub for functional brain networks. *Journal of Neuroscience*, *37*(23), 5594-5607.

Krauth, A., Blanc, R., Poveda, A., Jeanmonod, D., Morel, A., and Székely, G. (2010). A mean three-dimensional atlas of the human thalamus: generation from multiple histological data. *Neuroimage*, *49*(3), 2053-2062.